# Engineering programmable material-to-cell pathways via synthetic notch receptors to spatially control differentiation in multicellular constructs

Mher Garibyan [1,2,3,10], Tyler Hoffman [4,10], Thijs Makaske [1,2,8], Stephanie K. Do [3], Yifan Wu[4], Brian A. Williams[5], Alexander R. March [1,2], Nathan Cho[3], Nicolas Pedroncelli[4], Ricardo Espinosa Lima[4], Jennifer Soto[4], Brooke Jackson[4], Jeffrey W. Santoso [3], Ali Khademhosseini[4,9], Matt Thomson [5], Song Li [4,6,7], Megan L. McCain [1,3] ✉ & Leonardo Morsut [1,2,3] ✉

Synthetic Notch (synNotch) receptors are genetically encoded, modular synthetic receptors that enable mammalian cells to detect environmental signals and respond by activating user-prescribed transcriptional programs. Although some materials have been modified to present synNotch ligands with coarse spatial control, applications in tissue engineering generally require extracellular matrix (ECM)-derived scaffolds and/or finer spatial positioning of multiple ligands. Thus, we develop here a suite of materials that activate synNotch receptors for generalizable engineering of material-to-cell signaling. We genetically and chemically fuse functional synNotch ligands to ECM proteins and ECM-derived materials. We also generate tissues with microscale precision over four distinct reporter phenotypes by culturing cells with two orthogonal synNotch programs on surfaces microcontact-printed with two synNotch ligands. Finally, we showcase applications in tissue engineering by co-transdifferentiating fibroblasts into skeletal muscle or endothelial cell precursors in user-defined micropatterns. These technologies provide avenues for spatially controlling cellular phenotypes in mammalian tissues.

A fundamental goal for the emerging area of synthetic morphogenesis and tissue engineering is the ability to design and spatially control gene expression patterns within a multicellular construct. Intricate patterns of gene expression control the proper organization and physiology of cells, tissues, and organs and are a hallmark of complex multicellular systems across the tree of life. Individual cells express genetic networks that drive or support cell fate commitment and functional behaviors, like motility and proliferation. During embryonic

[1]Department of Stem Cell Biology and Regenerative Medicine, Keck School of Medicine of USC, University of Southern California, Los Angeles, CA, USA. [2]Eli and Edythe Broad Center, University of Southern California, Los Angeles, CA 90033, USA. [3]Alfred E. Mann Department of Biomedical Engineering, USC Viterbi School of Engineering, University of Southern California, Los Angeles, CA, USA. [4]Department of Bioengineering, University of California Los Angeles, Los Angeles, CA, USA. [5]Division of Biology and Biological Engineering, California Institute of Technology, Pasadena, CA, USA. [6]Broad Stem Cell Center, University of California, Los Angeles, Los Angeles, CA 90095, USA. [7]Jonsson Comprehensive Cancer Center, University of California, Los Angeles, Los Angeles, CA 90095, USA. [8]Present address: Utrecht University in the lab of Prof. Dr. Lukas Kapitein, Los Angeles, CA 90024, USA. [9]Present address: Terasaki Institute for Biomedical Innovation (TIBI), Los Angeles, CA 90024, USA. [10]These authors contributed equally: Mher Garibyan, Tyler Hoffman. ✉e-mail: mlmccain@usc.edu; morsut@usc.edu

development, initially, uniform cell ensembles activate genetic networks in designated spatial regions to generate tissues with distinct geometrical patterns. The spatial organization of cells within a tissue endows them to coordinate and accomplish complex functions, such as absorption or contractility. In vivo, spatial domains of gene expression are driven by genetically encoded communication networks involving intracellular[1,2], inter-cellular[3–7], and cell-to-extracellular matrix (ECM) and ECM-to-cell[8–10] components. Several of these networks are active in organoids in vitro, which self-organize and replicate select microscale architectural features similar to native tissues. However, genetic networks in organoids are spatially activated in an autonomous way and some genetic networks fail to activate at all, leading to heterogeneity and stunted tissue structures. Because self-organization is convoluted with differentiation and other complex cell behaviors, in vitro methods for arbitrarily engineering and interrogating spatial gene expression patterns and their impact would augment our understanding of biological systems[11–14]. Advanced technologies for spatially controlling gene expression would also enable tissues to be engineered with user-defined cellular compositions and geometries, which would be impactful for the fields of regenerative medicine, Organs on Chips, and lab-grown protein-rich food sources[15–18].

Classically, tissue engineers have focused on influencing cell differentiation and behavior by engaging endogenous cell surface receptors. For example, natural ligands, such as extracellular matrix proteins, can be presented to cells in user-defined spatial arrangements via microfabricated biomaterials to control adhesion, alignment, or differentiation[19–21]. Because these approaches rely on the engagement of endogenous receptors, such as integrins, stereotyped and often complex behaviors are activated in responding cells. However, with these approaches, users are confined to the limited library of endogenous ligands and receptors and their pre-existing downstream pathways, many of which are multifaceted with ambiguous outcomes. Recently, synthetic receptors have been developed that endow cells with orthogonal, customizable signaling capabilities[22]. Thus, we reasoned that these receptors could be leveraged to spatially control gene expression patterns in engineered tissues with more precision than endogenous receptors. Specifically, we turned to a class of synthetic receptors based on native Notch signaling, named synthetic Notch or synNotch[23]. SynNotch are a class of synthetic receptors composed of chimeric protein domains: an antibody-based binding extracellular domain (e.g. anti-GFP nanobody), the Notch juxtamembrane and transmembrane domains, and orthogonal transcription factors (e.g. Gal4) as the intracellular domain. SynNotch receptors have many desirable features that could be exploited to spatially control gene expression: (i) the receptor is not activated by soluble factors; (ii) the ligand is customizable and can be an orthogonal inert molecule, such as a fluorescent protein (e.g. GFP); (iii) receptor activation can drive customizable cellular responses, such as differentiation, when combined with complementary genetically engineered cassettes.

SynNotch has previously been used to generate spatial patterns of gene expression in 2-D (concentric rings[23]) and in 3-D (polarized and layered spheroids[24]) by using neighboring cells (i.e., sender cells) to present synthetic ligands to cells expressing synNotch (i.e., receiver cells). Cellular ligand presentation, however, has the disadvantage that controlling the geometry of synthetic ligands necessitates controlling the location of sender cells, making the problem circular. Evidence suggests that a pulling force between sender and receiver cells is necessary to initiate signal transduction in the receiving cell, similar to endogenous Notch receptors. Due to this feature, synNotch has also been activated by synthetic ligands passively adsorbed onto cell culture surfaces[23], tethered by DNA linkers to microbeads[25], and attached to atomic force microscopy probes[26]. More recently, an approach to specifically activate synNotch from culture surfaces was developed under the acronym MATRIX[27]. In this approach, surfaces are functionalized with antibodies (e.g. GFP-TRAP) that capture soluble synNotch ligands (e.g. GFP), which can then activate synNotch receptors (e.g. anti-GFP synNotch) in receiver cells to regulate CRISPR-based transcriptome modifiers, modulate inflammatory niches, and mediate stem-cell differentiation. Wedge-shaped culture inserts were also used to functionalize surfaces with coarse spatial control. However, whether synNotch ligands can be directly conjugated to a wider range of natural or synthetic biomaterials to activate synNotch, and whether this approach could be extended to pattern gene expression and/or differentiation and co-differentiation of multiple cell fates within the same culture with micron-scale precision, has not yet been shown.

Here, our objective was to develop generalizable, user-defined, material-to-cell pathways for spatially controlling genetic networks and differentiation in multicellular constructs via synNotch. We first show that we can activate synNotch with synthetic ligands (e.g., GFP) presented by materials that offer increasing degrees of spatial control: (i) genetically encoded, cell-produced ECM proteins (e.g., fibronectin-GFP fusions), (ii) ECM-derived hydrogels, and (iii) microcontact-printed culture surfaces. We also show that these approaches are generalizable to multiple synNotch receptors and can activate distinct synthetic pathways in cells with two synNotch receptors (i.e., dual-receiver cells). We then show that these approaches can be extended to spatially control patterns of gene expression and cell fate by transdifferentiating embryonic fibroblasts into either skeletal muscle precursors or endothelial cell precursors in tissue-relevant geometries. Finally, we demonstrate a method for spatially controlling the co-transdifferentiation of fibroblasts to one of two cell fates (endothelial cell precursors or skeletal muscle precursors) in a continuous tissue construct. This was achieved by generating dual-lineage fibroblasts expressing two independent synNotch receptors (one for endothelial transdifferentiation, and one for muscle transdifferentiation) and culturing these cells on a surface with the two synthetic cognate ligands patterned via a microfluidic device. These methods for generating spatial patterns of gene expression and cell fate add a powerful and flexible functionality to the synthetic biology toolbox for controlling and investigating multicellular organization.

## Results
### Activation of synNotch by particles and cell-generated ECM
To evaluate the activation of synNotch receptors by synthetic ligands presented on materials, we first used a suspension of microparticles to present ligands semi-analogously to the presentation of ligands on the membranes of sender cells. We tethered GFP to carboxyl-modified microparticles of different diameters (2 μm–10 μm) using an EDC/NHS reaction, which enables different amounts of GFP to be loaded by simply adjusting the concentration of GFP in the conjugation reaction. We then added these microparticles to a monolayer of receiver fibroblasts (L929 cells) that were engineered to express an anti-GFP/tTA synNotch receptor that activates a mCherry reporter gene (Fig. 1A). As expected, mCherry fluorescence at 24 h post-seeding increased with the concentration of GFP loaded onto the microparticles for all particle diameters and was absent when cells were presented with unmodified particles (Figs. 1B, C and S1A, B). Importantly, 5 μm microparticles loaded with 500 or 1000 μg/mL GFP-induced mCherry in the receiver fibroblasts at a level similar to GFP-presenting sender cells co-cultured at a 1:1 ratio, indicating that synthetic ligands conjugated to microparticles can activate synNotch receptors to a similar extent as synthetic ligands presented by sender cells.

We next asked if synNotch receptors could be activated by synthetic ligands presented on ECM fibers produced by cells. Thus, we genetically engineered mouse embryonic fibroblasts (3T3 cells) to produce a fusion protein of fibronectin and GFP (FN-GFP[28]). These cells were also engineered to express a far-red fluorescent nuclear reporter protein. We hypothesized that these FN-GFP-sender cells would deposit an ECM containing synthetic ligands that would signal to

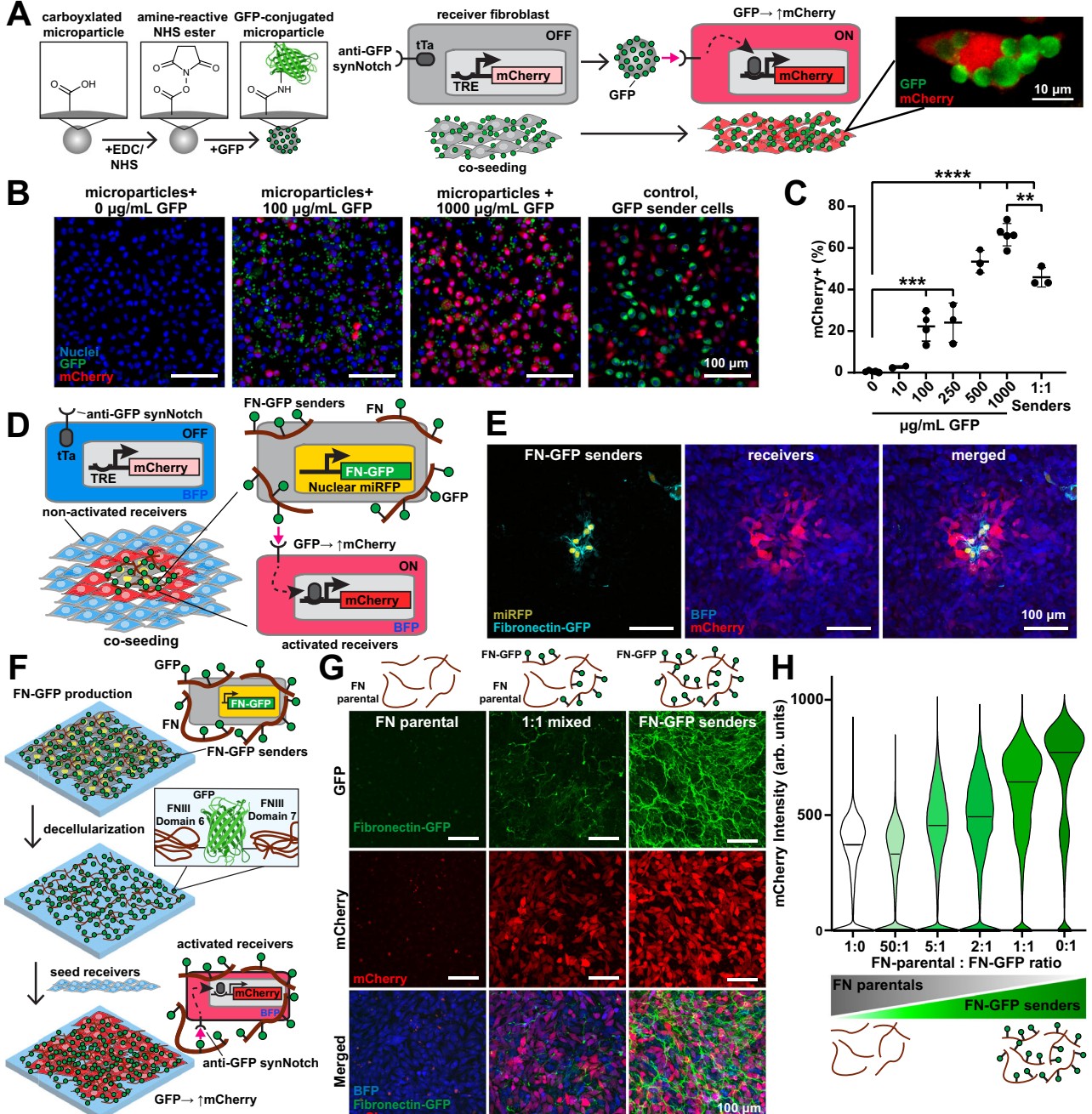

**Fig. 1 | Microparticle-conjugated GFP and cell-deposited fibronectin-GFP ligands activate reporter transgenes via synNotch in receiver fibroblasts.**
**A** Left: Schematic of GFP conjugation to microparticles and co-culture with receiver fibroblasts with anti-GFP/tTa synNotch that activates mCherry. Right: Fluorescence microscopy image of receiver fibroblast activated by GFP microparticles.
**B** Fluorescence microscopy images of receiver fibroblasts cultured with 5 µm microparticles conjugated with 0, 100, and 1000 µg/mL of GFP or GFP-sender cells for 1 day. Scale bars, 100 µm. **C** Percent of mCherry-expressing receiver fibroblasts quantified by image analysis following 24-h co-culture with GFP microparticles or GFP-sender cells. Data represent mean ± s.d. From left to right, $n = 5, 2, 4, 3, 3, 5, 3$ biological replicates. $p = 0.0012(**)$, $p = 0.0003(***)$, $p < 0.0001(****)$ determined via one-way ANOVA and Tukey's test. **D** Schematic of fibroblasts with miRFP nuclear tag producing fibronectin-GFP (FN-GFP) co-cultured with anti-GFP/tTa snyNotch receiver fibroblasts that activate mCherry and constitutively express BFP.

**E** Fluorescence microscopy images of FN-GFP senders and receiver fibroblasts co-cultured for 3 days. Scale bars, 100 µm. The experiment was performed three times with similar results. **F** Schematic of FN-GFP deposition by FN-GFP-sender cells, decellularization, and reseeding with receiver fibroblasts. **G** Top row: Schematics of decellularized extracellular matrix (ECM). FN is produced by parental fibroblasts (FN parental) and/or FN-GFP senders cultured at different ratios (1:0, 1:1, or 0:1) for 8 days prior to decellularization. Bottom: Fluorescence microscopy images of receiver fibroblasts cultured for 2 days on corresponding decellularized ECM. Scale bars, 100 µm. **H** Flow cytometry quantification of mCherry expression in receiver fibroblasts cultured for 2 days on decellularized ECM prepared by FN-parentals and/or FN-GFP fibroblasts. Data represent the distribution of individual cell intensity and median value from $n = 1$ biological replicate. The experiment was performed three times with similar results. Source data are provided as a Source Data file.

receiver cells. To test this, we co-cultured a low amount of FN-GFP-sender cells alongside receiver fibroblasts expressing anti-GFP/tTA synNotch receptors that activate mCherry (Fig. 1D). At 72 h after seeding, we observed mCherry expression only in receiver cells near to FN-GFP-sender cells (Figs. 1E and S1C), indicating that the anti-GFP synNotch receptor is locally activated in response to FN-GFP embedded in the ECM.

We then tested if cell-deposited FN-GFP matrices can activate receiver cells after the sender cells are removed. To do so, we cultured FN-GFP-sender cells as a monolayer for 8 days and subsequently performed decellularization to remove all cellular components while preserving the ECM (Figs. 1F and S1D). Receiver cells cultured on the decellularized matrices for 48 h expressed mCherry, indicating that synthetic ligands embedded in the ECM remained functional through the decellularization process. To tune the level of synNotch receptor activation by decellularized matrices, we co-cultured FN-GFP-sender cells with the unmodified parental 3T3 fibroblasts at various ratios. We similarly decellularized the co-cultured tissues and then seeded the decellularized matrices with receiver cells. mCherry intensity scaled with the ratio of parental cells to FN-GFP-sender cells in the original tissue (Fig. 1G, H), demonstrating tunability of activation of synNotch via cell-produced ECM fibers.

One advantage of synthetic receptors is that they can be engineered to both recognize distinct input ligands and drive distinct cellular responses. This feature has been used to generate a library of orthogonal synNotch receptors and pathways that function independently from each other and from endogenous receptors and pathways[23,24,29,30]. For example, synNotch receptors have been developed to recognize mCherry as its ligand[29]. To test if activation of synNotch receptors by matrix-presented synthetic ligands is generalizable to other ligand-receptor pairs, we generated FN-mCherry sender cells as well as corresponding receiver cells with anti-mCherry synNotch/Gal4 receptors that induce a BFP reporter gene upon activation. Similar to FN-GFP-sender cells, FN-mCherry sender cells activate receiver cells in co-culture and upon decellularization (Fig. S1E–H). We also observed that anti-mCherry receiver cells were activated by FN-mCherry decellularized matrices but not by FN-GFP decellularized matrices, illustrating the orthogonality of receptor activation by matrix-presented synthetic ligands (Fig. S1I, J). Overall, these data demonstrate that synNotch receptors can be robustly, tunably, and modularly activated by ligands presented on cell-produced ECM fibers.

## Activation of synNotch by hydrogels in 2- and 3-dimensions

To improve user control and tunability, we next tested if synNotch could be activated by ligands presented on purified ECM fibers processed into hydrogel biomaterials. As a first step, we attempted to activate synNotch receptors in cells cultured on the surface of matrix-derived hydrogels. Based on our previous protocols[31,32], we fabricated slabs of gelatin hydrogels enzymatically crosslinked with transglutaminase, an enzyme that cross-links glutamine and lysine residues[33]. We next sought to conjugate GFP onto the hydrogel surface with transglutaminase by adapting methods for conjugating laminin onto gelatin[34]. However, GFP is weakly reactive with transglutaminase because the glutamine and lysine residues of globular proteins are relatively inaccessible[35,36]. Thus, we synthesized GFP with a short C terminus LACE peptide tag (GFP-LACE) to provide accessible lysine residues[37]. We then treated gelatin hydrogels with a solution of GFP-LACE and transglutaminase to conjugate GFP onto the surface (Fig. 2A). When receiver cells with anti-GFP synNotch/tTA receptors that activate mCherry were cultured on the GFP-gelatin hydrogels, mCherry intensity increased in a GFP dose-dependent manner (Fig. 2B, C). Thus, synNotch receptors can be activated by synthetic ligands presented on the surface of matrix-derived hydrogels.

Next, we attempted to activate synNotch receptors in cells embedded in matrix-derived hydrogels that present synthetic ligands.

To do so, we developed a click chemistry method to conjugate synthetic ligands to hydrogels (Fig. 2D). Briefly, GFP was modified with trans-Cyclooctene (TCO) NHS ester to generate GFP-TCO moieties. In parallel, the gelatin polymer was modified with methacrylate (MA) groups for photo-cross-linking and methyltetrazine (mTz) to generate GelMA-mTz (Fig. S2A, B). These coordinated substitutions enable facile conjugation of TCO-modified protein ligands to the mTz-modified hydrogel polymer backbone via rapid click reaction after mixing[38,39] (Fig. S2C). Combining GelMA-mTz with GFP-TCO generated GelMA-GFP, which could then be photocrosslinked into a hydrogel that demonstrated retention of the GFP ligand for at least seven days (Fig. S2D, E). We then embedded anti-GFP receiver fibroblasts in GelMA-GFP hydrogels via photocrosslinking. mCherry expression in receiver cells significantly increased in GelMA-GFP hydrogels but not in unmodified GelMA hydrogels, with approximately 70% of receiver cells within GelMA-GFP demonstrating sustained activation for up to 7 days (Figs. 2E, F and S2E–H). Cell viability was also maintained for encapsulated cells (Fig. S2I). In contrast, when we attempted to activate synNotch receiver cells via sender cells co-embedded in a GelMA hydrogel, only approximately 30% of receiver cells were activated (Fig. S3A, B).

To demonstrate spatial confinement of activation, we next encapsulated receiver fibroblasts via manual pipetting in a biphasic GelMA hydrogel, where only half of the hydrogel contained GFP. Due to the covalent linkage between GFP and GelMA, the spatial position of GFP was maintained over time and the GFP did not diffuse through the hydrogel (Fig. S3C). As shown in Fig. 2G–I, mCherry activation was similarly spatially restricted to the GelMA-GFP region over time, demonstrating that the GFP ligand conjugated to the hydrogel-activated synNotch only in the regions where it was originally positioned. To validate the generalizability of this method, we also engineered fibrinogen-mCherry constructs via a similar click chemistry approach and then embedded anti-mCherry/Gal4 synNotch receiver cells that activate BFP in these hydrogels (Fig. S3D). As shown in Fig. S3E, receiver cells activated only in fibrinogen-mCherry hydrogels but not unmodified fibrinogen hydrogels. Collectively, these results demonstrate that matrix-derived hydrogels can be covalently conjugated with synthetic ligands to generate 2- or 3-dimensional materials capable of locally activating receiver cells.

## Spatial activation of synNotch via microcontact printing

Our next goal was to dictate synNotch activation patterns within multicellular tissue constructs at a spatial resolution similar to the cellular length scale. To achieve this, we adapted microcontact printing techniques designed to transfer microscale patterns of proteins (classically ECM proteins) onto culture surfaces[40,41]. Our goal was to microcontact print GFP onto uniformly cell-adhesive surfaces. To achieve this, we treated PDMS-coated coverslips with APTES and glutaraldehyde to induce covalent bonding of proteins[42] and then coated the surface with fibronectin for uniform cell adhesion. To optimize the transfer of GFP onto the fibronectin layer, we created simple, featureless PDMS stamps by cutting cylinders from PDMS using a biopsy punch. We coated and incubated these stamps with 0–200 µg/mL GFP and then inverted them onto fibronectin-coated coverslips. Finally, we seeded coverslips with receiver cells expressing anti-GFP/tTA synNotch receptors that activate a mCherry reporter. These cells formed a confluent monolayer and demonstrated a GFP dose-dependent increase in mCherry fluorescence that saturated at approximately 100 µg/mL GFP (Fig. S4A, B), indicating that surfaces dual-functionalized with fibronectin and GFP maintained cell adhesion and activated synNotch.

To induce activation of synNotch in small groups of cells within a multicellular tissue, we next developed an approach to microcontact print arrays of GFP squares with features ranging from 100 µm to 1 mm (Fig. 3A). PDMS stamps for microcontact printing

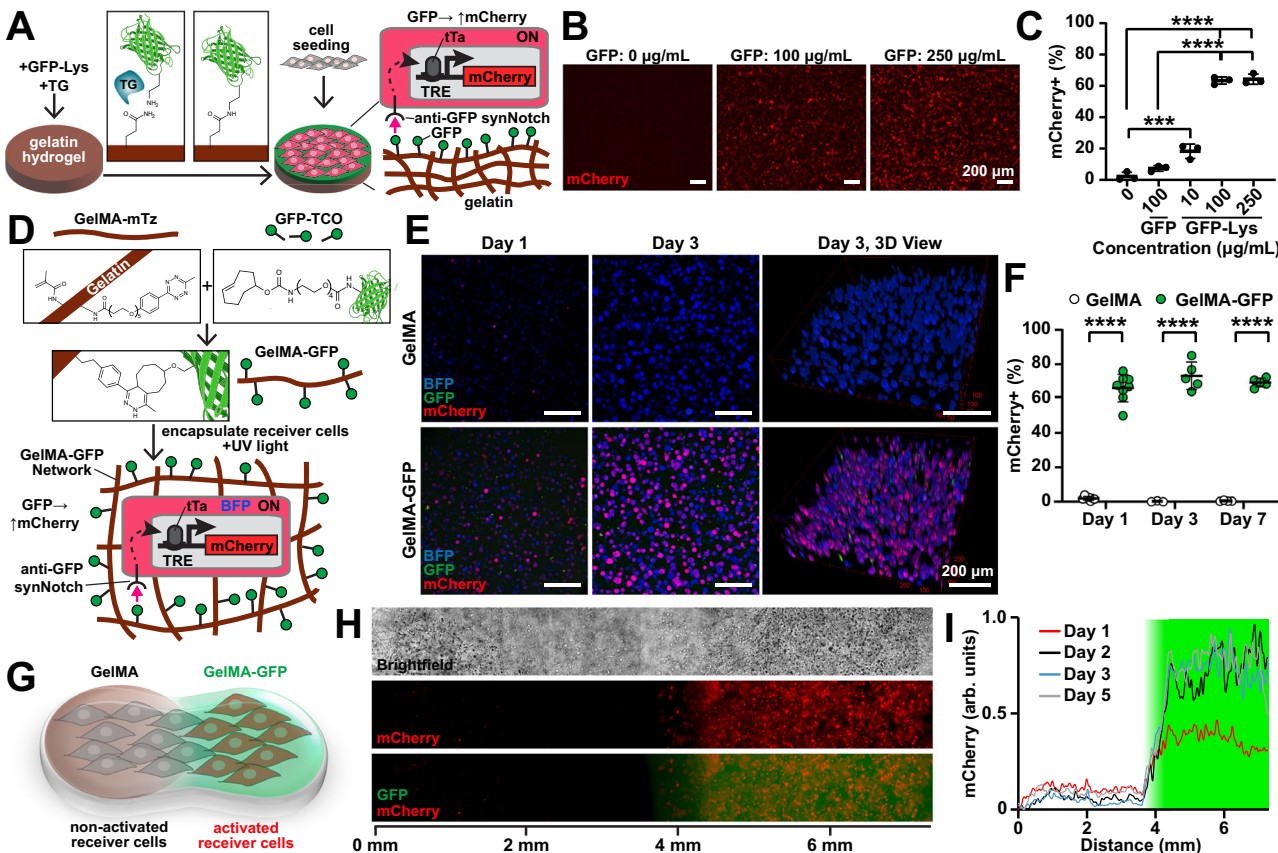

**Fig. 2 | Hydrogel-conjugated GFP ligands activate reporter transgenes via synNotch with coarse spatial control in receiver fibroblasts. A** Schematic of enzymatic conjugation of GFP-Lys via transglutaminase (TG) to the surface of a gelatin hydrogel to activate anti-GFP/tTa synNotch that activates mCherry in receiver fibroblasts. **B** Fluorescence microscopy images of receiver fibroblasts cultured on gelatin hydrogels conjugated with GFP-Lys at 0, 100, or 250 μg/mL. Scale bars, 200 μm. **C** Flow cytometry quantification of percent of receiver fibroblasts expressing mCherry following culture for three days on gelatin hydrogels conjugated with GFP or GFP-Lys at indicated concentrations. Data represent mean ± s.d., $n = 3$ biological replicates. $p = 0.0006(***)$, $p < 0.0001(****)$ determined via one-way ANOVA and Tukey's test. **D** Schematic of gelatin methacryloyl-methyltetrazine (GelMA-mTz) covalent conjugation to GFP-trans-cyclooctene (GFP-TCO) and subsequent photoencapsulation of receiver fibroblasts with UV light. **E** Z-projections and 3-D views of confocal microscopy images of receiver

fibroblasts encapsulated in GelMA-mTz hydrogels containing 0 (GelMA) or 50 μg/mL GFP-TCO (GelMA-GFP) at Day 1 and 3 of culture. Scale bars, 200 μm. See Fig. S2G for day 7. **F** Percent of mCherry-expressing receiver cells quantified by image analysis after 1, 3, and 7 days of culture within GelMA or GelMA-GFP hydrogels. Data represent mean ± s.d. From left to right, $n = 7, 8, 3, 5, 4, 5$ biological replicates. $p < 0.0001(****)$ determined via unpaired two-tailed $t$-test. **G** Schematic of encapsulation of receiver fibroblasts into biphasic GelMA/GelMA-GFP hydrogel generated by micropipetting. **H** Brightfield and fluorescence microscopy images of receiver fibroblasts in biphasic GelMA/GelMA-GFP hydrogel 5 days after encapsulation. **I** Profile plot of normalized mCherry intensity across the length of the hydrogel 1, 2, 3, and 5 days after encapsulation. The green shaded area indicates the GelMA-GFP region. Data are shown from one replicate. The experiment was performed three times with similar results. See Fig. S3C for quantification of the GFP signal. Source data are provided as a Source Data file.

are classically cast on silicon wafer templates fabricated by cleanroom-based photolithography[43]. However, this approach is not suitable for our feature sizes because they are large (100 μm–1 mm) relative to the height of the photoresist conventionally used for photolithography (1–10 μm). PDMS stamps with high feature-to-height ratios are susceptible to buckling and transfer of GFP outside the intended regions[43]. To overcome this, we used a digital light processing (DLP) 3-D printer to rapidly print templates with taller features in a photocrosslinkable resin. We first 3-D printed a template comprising an array of 100-μm-sided-squares with 100 μm interspaces, which is roughly the resolution limit of the 3-D printer. The height of the features was set as 100 μm to minimize buckling. As shown in Fig. 3B, PDMS stamps fabricated in this way could successfully transfer GFP onto covalently coated FN coverslips in the intended 100 μm × 100 μm pattern, demonstrating successful microcontact printing using PDMS stamps cast on 3-D-printed templates.

We next used these techniques to fabricate stamps and microcontact print arrays of GFP squares with sides ranging from 250 μm to

1000 μm and interspaces of 250 μm or 500 μm onto PDMS-coated coverslips pre-coated with fibronectin. The feature height for these stamps ranged from 100 μm to 500 μm, depending on square sizes and interspaces. Microcontact-printed surfaces were then seeded with receiver cells with anti-GFP/tTA synNotch receptors that activate a mCherry reporter (Fig. 3C). After two days, mCherry expression was detected within the multicellular tissue in patterns that overlapped with the original design to different extents, depending on the pattern (Figs. 3D, E and S4C, D). To quantify the spatial fidelity of synNotch activation, we calculated Pearson's correlation coefficient between the binary user-defined pattern and the mCherry images (Figs. 3F and S4E). As expected, the correlation coefficient was highest for tissues with the largest squares (500 μm sides) and largest interspaces (1000 μm). The correlation coefficient decreased as features and/or gaps decreased. However, for all tissues with square sizes and interspaces greater than 100 μm, the correlation coefficient between the mCherry image and the binary pattern was significantly higher compared to the correlation coefficient between the mCherry image and a scrambled binary pattern with the same number of white pixels. The correlation also

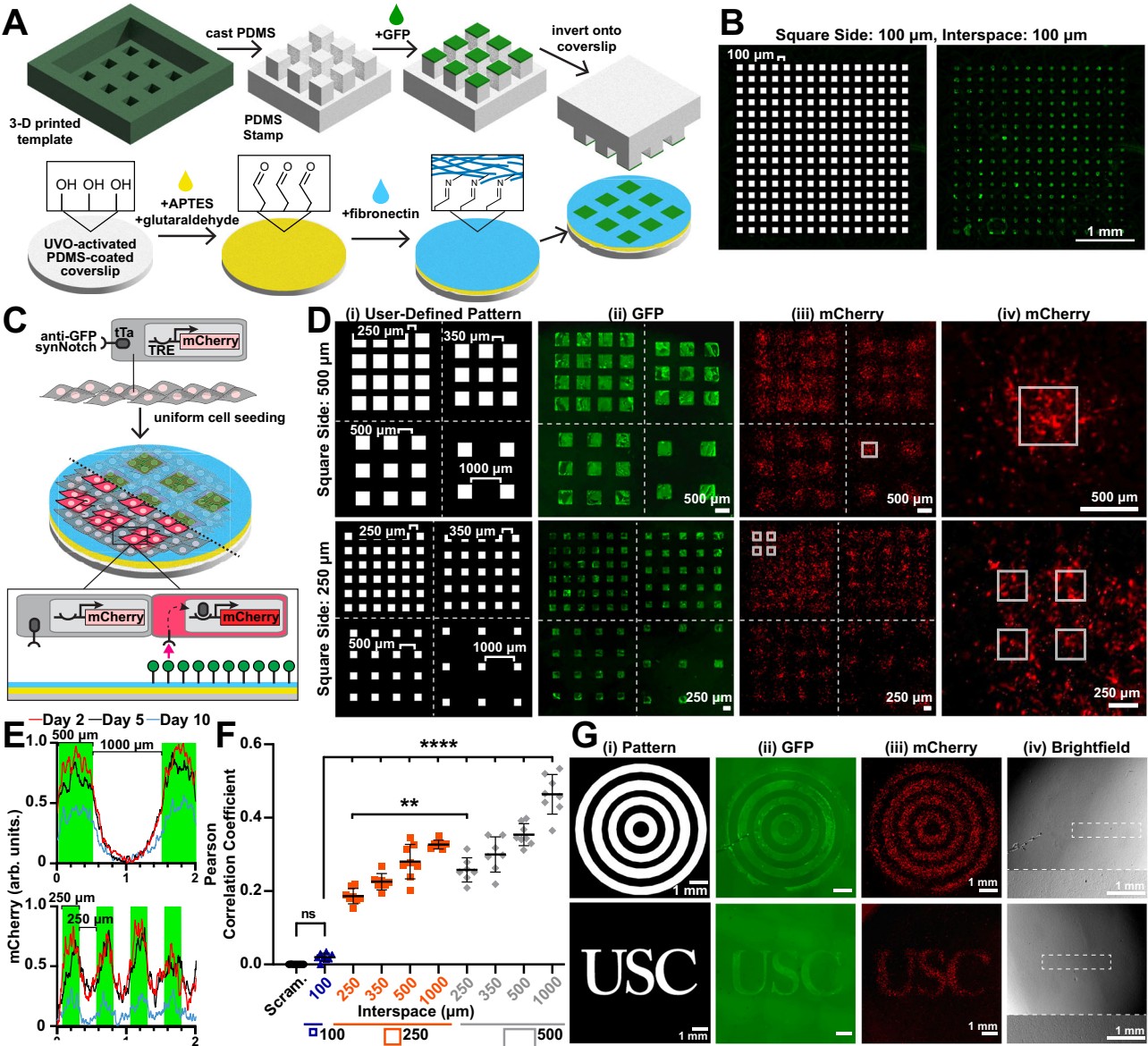

**Fig. 3 | Microcontact-printed patterns of GFP ligands activate reporter transgenes via synNotch with microscale spatial control in receiver fibroblasts.**
**A** Schematic of stamp and coverslip preparation and microcontact printing of GFP. **B** Pattern and corresponding fluorescence microscopy image of microcontact-printed GFP. **C** Schematic of anti-GFP/tTA synNotch receiver fibroblasts that activate mCherry cultured on substrates microcontact-printed with GFP. Half the tissue is transparent to visualize the underlying GFP pattern. **D** (i) Patterns and (ii) resulting fluorescence microscopy images of microcontact-printed GFP. (iii, iv) Fluorescence microscopy images of mCherry expressed by receiver cells cultured on GFP-patterned substrates for two days. Dotted white lines separate regions with different interspaces. Solid white boxes in (iii) and (iv) indicate the location of the printed GFP. **E** Profile plot of normalized mCherry intensity in receiver cells after indicated days of culture on 500-μm-sided GFP squares with 1000 μm interspaces (top) and 250-μm-sided GFP squares with 250 μm interspaces (bottom). Green indicates the locations of printed GFP. **F** Pearson correlation coefficients generated by comparing images of user-defined patterns to fluorescence microscopy images of mCherry intensity in receiver cells after two days of culture. A scrambled pattern obtained by scrambling the pixels of the user-defined pattern (see "Methods" section) was correlated to their respective mCherry fluorescence microscopy images as a negative control. Data represent mean ± s.d. From left to right, $n = 8, 8, 8, 7, 8, 7, 7, 7, 8, 8$ biological replicates. $p = 0.9697$ (not significant, ns), $p = 0.0029$(**), $p < 0.0001$(****) determined via one-way ANOVA and Tukey's test. **G** (i) Patterns and corresponding fluorescence microscopy images of (ii) microcontact-printed GFP and (iii) mCherry expression by receiver fibroblasts after two days of culture. (iv) Corresponding brightfield microscopy images. The region outlined by the dotted white rectangle is shown at a higher magnification at the bottom of the same image. Experiment performed three times with similar results. Scale bars, 1 mm. Source data are provided as a Source Data file.

decreased with time (Figs. 3E and S4D, E). Together, these data indicate that the minimum feature size for this approach is approximately 250 μm. Based on this conclusion, we designed other arbitrary patterns with minimal feature sizes of 250 μm, including concentric circles and letters, and qualitatively observed similar agreement between the binary pattern, GFP fluorescence, and mCherry fluorescence (Fig. 3G), demonstrating the versatility of pattern designs.

Our next goal was to scale up this approach to spatially activate multiple orthogonal genetic programs in the same multicellular tissue. Previous studies have demonstrated that two synNotch receptors can be integrated into a single dual-receiver cell[23]. Thus, we asked if culturing dual-receiver cells on a surface patterned with two synthetic ligands in distinct arrangements would generate a tissue with corresponding patterns of distinct genetic programs. We first generated a

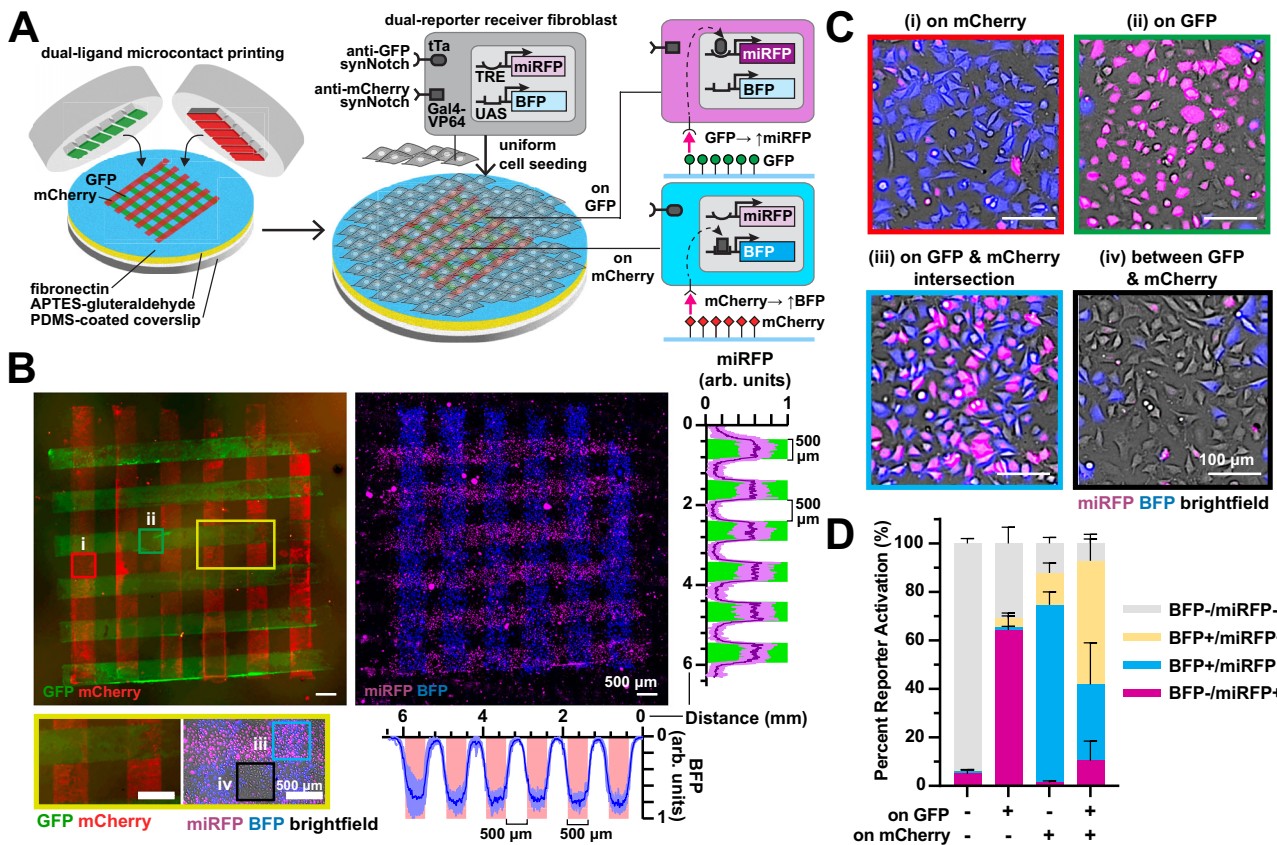

**Fig. 4 | Microcontact-printed GFP and mCherry ligands spatially activate distinct reporter genes via synNotch in dual-reporter receiver fibroblasts.**
**A** Schematic of dual-ligand microcontact printing and seeding of dual-receiver fibroblasts with anti-GFP/tTa synNotch that activates miRFP and orthogonal anti-mCherry/Gal4-VP64 synNotch that activates BFP. **B** Fluorescence microscopy images of microcontact-printed 500-µm-wide perpendicular lines of GFP and mCherry and corresponding miRFP and BFP expression by dual-receiver cells after one day of culture. Scale bars, 500 µm. Right: Profile plot of normalized miRFP intensity across the y-axis, green bars indicate regions patterned with GFP. The line represents mean ± s.d., $n = 7$ biological replicates. Below: Profile plot of normalized BFP intensity across the x-axis, red bars indicate regions patterned with mCherry. The line represents mean ± s.d., $n = 7$ biological replicates. The region outlined by the yellow rectangle is shown at a higher magnification below the image. Regions patterned with (i, red square) mCherry only, (ii, green square) GFP only, (iii, blue square) mCherry and GFP, and (iv, black square) neither mCherry nor GFP. **C** Higher magnification fluorescence microscopy images of regions shown in (**B**). Scale bars, 100 µm. **D** Percent of BFP and/or miRFP-expressing receiver cells after one day of culture on the indicated GFP/mCherry patterns, quantified by image analysis. $n = 4$ biological replicates, data represent mean ± s.d. See Fig. S5F, G for more data on this. Source data are provided as a Source Data file.

dual-receiver fibroblast cell line (L929) that harbors an anti-GFP/tTA synNotch receptor that activates a miRFP reporter and an anti-mCherry/Gal4 synNotch that activates a BFP reporter (Fig. S5A). To validate the responses to synthetic ligands of these cells, we seeded them on a culture surface adsorbed with a uniform layer of GFP, mCherry, or both. As shown in Fig. S5B, C, miRFP was expressed only on GFP surfaces and BFP was expressed only on mCherry surfaces, demonstrating orthogonal activation of the two pathways. We also observed that the anti-mCherry pathway activated with higher efficiency than the anti-GFP pathway, suggesting different levels of receptor activation. On surfaces with both GFP and mCherry, both miRFP and BFP were expressed, indicating activation of both pathways concurrently in the same cells. Next, to prototype the generation of spatial patterns of gene expression starting from a uniform population of dual-receiver cells, we adsorbed a GFP droplet in one corner of a culture surface and a mCherry droplet in the opposing corner. Dual-receiver cells cultured uniformly on the surface activated miRFP and BFP in a spatial pattern corresponding to the GFP and mCherry droplets, respectively (Fig. S5D, E), demonstrating macroscale spatial control over the activation of two synNotch pathways in one cell population.

Subsequently, to provide more precise spatial control over the patterns, we microcontact-printed an array of 500-µm-wide rows of GFP with 500 µm interspacing. We then stamped perpendicular mCherry rows by manually positioning the orientation of the stamp (Fig. 4A). When seeded with dual-receiver cells, we observed rows of miRFP-expressing cells perpendicular to rows of BFP-expressing cells (Figs. 4B and S5F), as expected. At the GFP and mCherry intersections, cells expressed both miRFP and BFP (Figs. 4C and S5G), indicating activation of both synNotch pathways, generating four reporter "states" for the initially uniform population of engineered cells (BFP−/miRFP−, BFP−/miRFP+, BFP+/miRFP−, BFP+/miRFP+) within the 1.5 mm² tissue. Additionally, we quantified the percent of BFP and miRFP expression in cells on different regions of the pattern with image analysis (Fig. 4D). Approximately 60–70% of dual-receiver cells on a region with a single ligand (GFP or mCherry) expressed the matching reporter (miRFP or BFP, respectively). On the GFP and mCherry intersections, approximately 50% of dual-receiver cells expressed both BFP and miRFP. These values were similar to the percent reporter activation measured by flow cytometry in dual-receiver fibroblasts cultured on surfaces uniformly adsorbed with one or both ligands (Fig. S5C). At the intersections, we also noticed a larger

proportion of BFP+/miRFP− cells compared to BFP−/miRFP+ cells, which is consistent with the higher efficiency of the anti-mCherry/BFP pathway observed in Fig. S5C. This may be due to differential signal transduction of receptor-ligand pairs, differential adsorption of the two ligands to the surface, the order of ligand printing, or other factors. Collectively, these data show that two independent synNotch genetic programs can be spatially controlled by culturing dual-receiver cells on user-defined patterns of the two synthetic ligands, generating a multicellular tissue with up to four spatially controlled reporter gene expression states.

## Spatial control of myoblast transdifferentiation

Beyond expression of fluorescent reporter proteins, synNotch receptors have also been used to activate transgenes that control cell phenotypes or behaviors via overexpression of transcription factors, such as Snail for epithelial to mesenchymal transitions or myoD for transdifferentiation of fibroblasts to skeletal muscle precursors[23]. Thus, we next tested if synthetic ligands presented by materials could drive overexpression of functional transcription factors that induce transdifferentiation. We first generated a receiver fibroblast cell line (C3H) expressing an anti-GFP/tTA synNotch receptor that activates myoD (Fig. 5A). When these receiver cells were cultured on surfaces uniformly printed with GFP, they transdifferentiated to multinucleated, α-actinin-positive myotubes (Fig. 5B). To further characterize changes in phenotype, we performed bulk RNA sequencing on unmodified C3H fibroblasts, receiver cells cultured on surfaces with or without GFP, and C2C12 myotubes. We observed that culturing receiver cells on GFP surfaces led to 3064 differentially expressed genes. According to hierarchical clustering, receiver cells on GFP were most similar to C2C12 myotubes (Fig. 5C). Receiver cells on GFP also overexpressed several muscle-specific genes, such as *Myh2*, *Myh4*, and *Ttn*, and down-regulated expression of fibroblast genes, such as *Col1a1* and *Pdgfrb* (Fig. 5D). GO-term analysis indicated that several pathways related to muscle development and differentiation were enriched in receiver cells on surfaces with GFP compared to without GFP (Fig. 5E). In contrast, receiver cells expressing an anti-GFP/tTA synNotch receptor that activates mCherry did not over-express muscle-specific genes or pathways, and only led to 33 differentially expressed genes, when cultured on surfaces with or without GFP (Fig. S6A). Together, these data indicate that surfaces with GFP specifically induced the transdifferentiation of receiver cells expressing an anti-GFP synNotch receptor that activates MyoD to myogenic precursors.

Our next goal was to combine the synNotch receptor technology with surface micropatterning to engineer aligned muscle tissue. Previous studies have shown that micromolded gelatin hydrogels are favorable for myotube adhesion and alignment[32,44]. Thus, we asked if this type of surface could be used to both transdifferentiate and align synNotch-induced myotubes. We constructed gelatin hydrogels that are either isotropic or micromolded with 10 μm ridges separated by 10 μm spacing and then enzymatically conjugated GFP to the surface using the procedure described above (see Fig. 2A). Receiver cells cultured on GFP hydrogels transdifferentiated to α-actinin-positive myotubes, independent of surface topography, and receiver cells consistently aligned to micromolded ridges (Figs. 5F–H and S6B, C), independent of activation state. However, only receiver cells cultured on micromolded GFP hydrogels fused into aligned myotubes, demonstrating that transdifferentiation and cell alignment were controlled independently. We did observe a slight but non-significant increase in nuclei alignment for cells cultured on micromolded gelatin hydrogels with GFP compared to without GFP (Fig. S6C), possibly because cell fusion induced by MyoD caused a modest improvement in cell alignment.

Another approach for engineering aligned muscle tissues is to culture muscle cells on lines of matrix proteins microcontact-printed on otherwise non-adherent surfaces[45]. We tested if this approach was compatible with synNotch by microcontact printing lines using a mixture of fibronectin and GFP. When the same receiver cells were cultured on these surfaces, they transdifferentiated into aligned myotubes (Fig. S6D), indicating that microcontact printing matrix proteins and synthetic ligands can also be used to both control tissue architecture and transdifferentiation.

In the approaches described above, a population of fibroblasts was uniformly transdifferentiated to myoblasts. Our next goal was to selectively transdifferentiate fibroblasts to myoblasts in a spatially controlled manner as a first step towards generating tissues with multiple distinct cell types arranged in prescribed patterns. To achieve this, we used the approach described above (Fig. 3) to microcontact print rows of GFP on fibronectin-coated surfaces. To test if we could achieve spatially controlled differentiation, we printed thin or thick, curved or straight, rows and then seeded the printed surfaces with fibroblasts harboring an anti-GFP synNotch receptor that activates myoD (Fig. 5I). After three days, we fixed and stained tissues for α-actinin and quantified the myogenic index on and off the pattern by using the user-defined pattern as a mask (Figs. 5I, J and S6E–I). The myogenic index was significantly higher on GFP compared to off GFP for all geometries, demonstrating local geometric control of transdifferentiation. We also quantified the orientation order parameter as a proxy for alignment and observed higher alignment for tissues only on the straight 200 μm rows compared to isotropic tissues (Fig. 5K). Thus, we can selectively transdifferentiate fibroblasts to myoblasts in a geometrically prescribed way while also controlling the global alignment of the tissue, demonstrating that we can separately and concurrently control local differentiation and tissue architecture.

## Spatial control of endothelial transdifferentiation

To exploit the modularity of this technology, we next tested if transdifferentiation to another cell fate could be activated by a similar approach. Due to the universal need for vascularization in engineered tissue constructs, including muscle, we focused on transdifferentiating fibroblasts into endothelial cell precursors, which was previously shown via doxycycline-inducible overexpression of the master transcription factors ETV2[46,47]. Thus, we generated fibroblast receiver cells engineered with an anti-mCherry/Gal4 synNotch receptor that activates an ETV2-BFP cassette (Fig. 6A). We then passively adsorbed mCherry onto culture surfaces, cultured receiver cells on them for three days, and fixed and stained the cells for endothelial cell precursor markers. As shown in Figs. 6B, C and S7A–C, the fibroblasts transdifferentiated to VEGFR2-positive endothelial precursors that also expressed VE-cadherin on their membrane. CDH5 (VE-Cadherin), a later-stage endothelial marker, was also detected at the protein level with flow cytometry (Fig. S7D, E). We also evaluated the differentiation trajectory by performing bulk RNA sequencing of receiver cells cultured on surfaces with or without mCherry, unmodified C3H fibroblasts, and an endothelial cell line (Bend.3) a positive control. Culturing receiver cells on mCherry surfaces led to 3022 differentially expressed genes compared to culturing on surfaces without mCherry. Receiver cells cultured on mCherry preferentially clustered with Bend.3 endothelial cell line (Fig. 6D) and overexpressed endothelial-related genes, such as *KDR* and *CDH5*, compared to cells cultured on surfaces without mCherry (Fig. 6E). These data demonstrate that receiver cells expressing an anti-mCherry synNotch receptor that activates ETV2 transdifferentiated to endothelial cell precursors via mCherry adsorbed on a culture surface.

To test if we can also control the geometry of transdifferentiation for the endothelial lineage, we generated uniformly adhesive surfaces and then microcontact-printed mCherry in varying designs. Fibroblast receivers are activated by mCherry and express VEGFR2 based on the original ligand patterning, where we evaluated a pattern with 500 μm rows (Fig. 6F, G). We also designed a pattern to replicate a branching network structure typical of vascular beds[48] and showed the formation

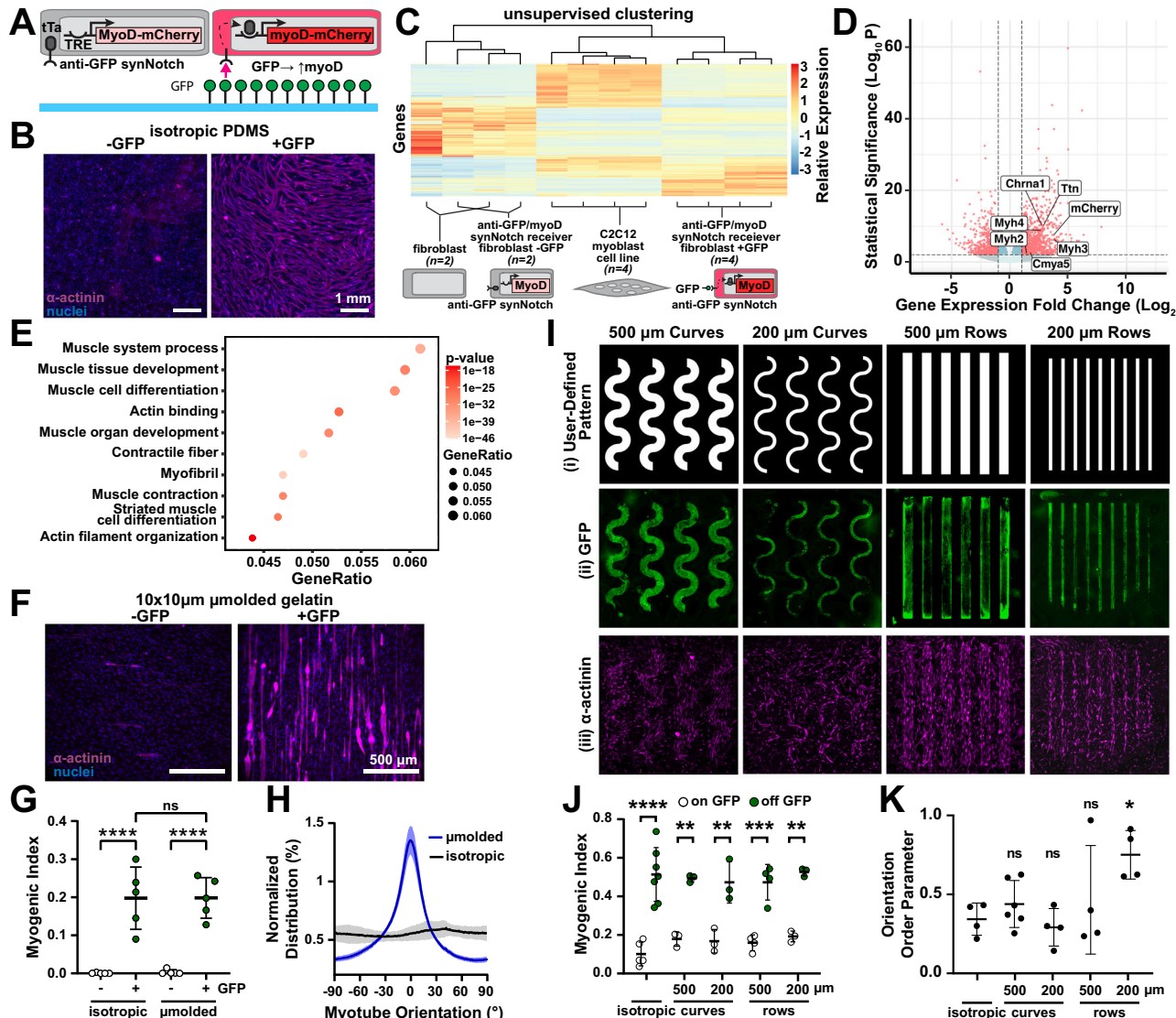

**Fig. 5 | Micropatterned GFP ligands spatially activate MyoD and initiate myotube differentiation in receiver fibroblasts via synNotch. A** Schematic of receiver fibroblasts with anti-GFP/tTa synNotch that activates mCherry and myoD on substrates with GFP. **B** Fluorescence microscopy images of receiver fibroblasts cultured on PDMS-coated coverslips with (+GFP) or without (−GFP) microcontact-printed GFP for three days and stained for sarcomeric α-actinin (purple) and DAPI (blue). **C** Heatmap and hierarchical clustering of gene expression, measured by bulk RNA sequencing, for indicated cell types and substrates. n=biological replicates. **D** Volcano plot and **E** gene ontology analysis of differentially expressed genes in receiver fibroblasts on +GFP ($n = 4$ biological replicates) compared to −GFP ($n = 2$ biological replicates). **F** Fluorescence microscopy images of receiver fibroblasts after four days on micromolded gelatin hydrogels with (+GFP) or without (−GFP) conjugated GFP, stained for sarcomeric α-actinin (purple) and DAPI (blue). **G** Myogenic index for receiver fibroblasts on indicated gelatin hydrogel substrates

for four days. Data represent mean ± s.d, $n = 5$ biological replicates, $p > 0.9990$ (not significant, ns), $p < 0.0001$(****) determined via one-way ANOVA and Tukey's test. **H** Myotube orientation for receiver fibroblasts cultured on indicated +GFP-gelatin hydrogels. Solid line and shading represent mean and s.d. of five images from one tissue. Experiment performed in five tissues with similar results. **I** (i) Patterns and (ii) resulting fluorescence microscopy images of microcontact-printed GFP and (iii) receiver fibroblasts cultured for four days, stained for sarcomeric α-actinin (purple) and DAPI (blue). **J** Myogenic index for receiver fibroblasts cultured on indicated regions for three days. Data represent mean ± s.d. From left to right, $n = 5, 7, 3, 3, 3, 3, 4, 4, 3, 3$; $p = <0.0001$ (****), 0.003 (**), 0.0043 (**), 0.0005 (***), 0.0014 (**) determined via one-way ANOVA and Tukey's test. **K** Orientation order parameter of α-actinin immunosignal. Data represent mean ± s.d. From left to right, $n = 4, 6, 4, 4, 4$; $p = 0.9317$ (ns), 0.9948 (ns), 0.8879 (ns), 0.0497 (*) compared to isotropic via one-way ANOVA and Tukey's test. Source data are provided as a Source Data file.

of a tissue consisting of activated cells in the corresponding pattern surrounded by a uniform layer of fibroblasts (Figs. 6H and S7F). Thus, similar to the myogenic synNotch cells, microcontact-printed ligands can activate SynNotch-induced transdifferentiation to endothelial precursors with spatial control.

With two differentiating synNotch receiver cell lines now in hand (myogenic, endothelial), we next compared their transcriptome as a function of surface-presented ligands and to positive and negative control cells. Specifically, we performed PCA among receiver fibroblasts induced or not induced to transdifferentiate

towards the endothelial or myogenic lineage by mCherry or GFP respectively, receiver fibroblasts induced or not induced to express a fluorescent reporter, unmodified parental fibroblasts, and cell type-specific cell lines (C2C12 myoblasts, Bend.3 endothelial cells). As shown in Fig. S7G, the respective receptor-ligand pair in the transdifferentiating receiver cells pushed the cells away from the unmodified parental fibroblasts and towards the expected muscle or endothelial cell line. Receiver cells expressing synNotch that activate fluorescent proteins also clustered with unmodified parental fibroblasts in both the presence and absence of their respective

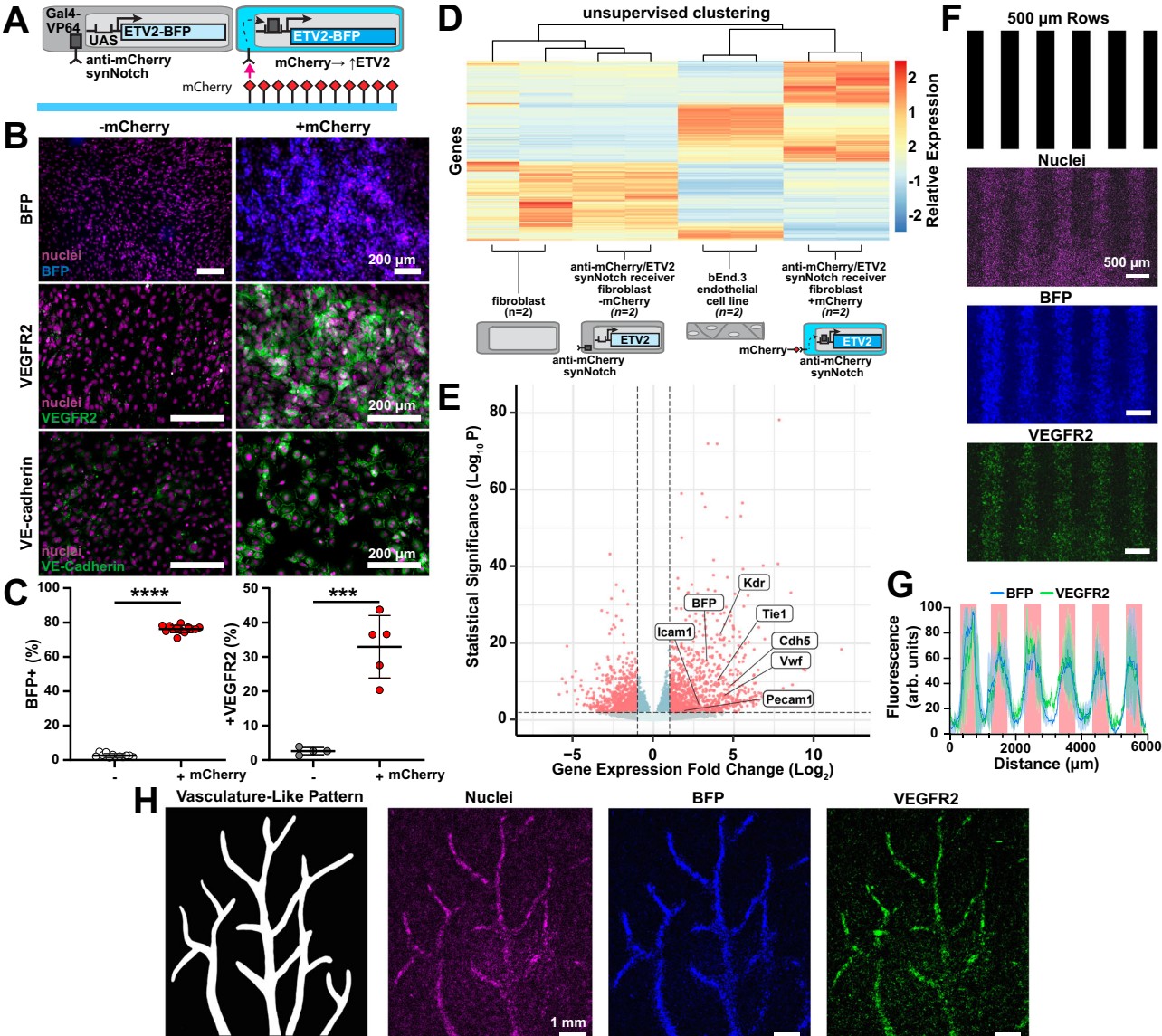

**Fig. 6 | Micropatterned mCherry ligands spatially activate ETV2 and initiate endothelial differentiation in receiver fibroblasts via synNotch. A** Schematic of receiver fibroblasts with anti-mCherry/Gal4 synNotch that activates BFP and ETV2 when cultured on substrates with mCherry. **B** Fluorescence microscopy images of receiver fibroblasts cultured for three days on substrates with (+mCherry) or without (−mCherry) mCherry coating and immunostained for VEGFR2 (middle, green) and VE-cadherin (bottom, green). BFP reporter (top, blue) and nuclei (purple, all) also shown. Scale bars, 200 μm. **C** Percent of receiver fibroblasts expressing BFP (left) and VEGFR2 (right) after 3 days of culture on indicated substrates, quantified with flow cytometry. Data represent mean ± s.d, BFP $n = 12$ (−mCherry) $n = 13$ (+mCherry), VEGFR2 $n = 4$ (−mCherry) $n = 5$ (+mCherry) biological replicates, $p = 0.003(***)$, $p < 0.0001(****)$ determined via unpaired two-tailed $t$-test. **D** Heatmap and hierarchical clustering of gene expression, measured by bulk RNA sequencing, for the indicated cell types on the indicated substrates. $n = 2$ biological

replicates. **E** Volcano plot of differentially expressed genes in receiver fibroblasts cultured on substrates with mCherry compared to without mCherry. $n = 2$ biological replicates. **F** Pattern used to make stamps for microcontact printing rows of mCherry and resulting fluorescence microscopy images of receiver fibroblasts cultured on substrates for three days and immunostained for VEGFR2 (green). BFP reporter (blue) and nuclei (purple) also shown. Scale bars are 500 μm. **G** Profile plot of normalized BFP and VEGFR2 intensity across the y-axis, red bars indicate regions patterned with mCherry. The line represents mean ± s.d, $n = 2$ biological replicates. **H** Pattern used to make stamps for microcontact printing mCherry into a vasculature-like pattern and resulting fluorescence microscopy images of receiver fibroblasts cultured on substrates for three days and immunostained for VEGFR2 (green). BFP reporter (blue) and nuclei (purple) also shown. Scale bars are 1 mm. Source data are provided as a Source Data file.

ligand, as expected. We also noticed that receiver cells expressing anti-GFP synNotch that activates MyoD had a significant shift from the negative control cells towards C2C12 cells, even in the absence of GFP, suggesting some basal level of non-specific activation of the receptor. This was not observed for receiver cells expressing the anti-mCherry synNotch that activates ETV2, suggesting that non-specific activation could depend on the nature of the transdifferentiation factors or on the expression level of the receptors and transgenes of the synthetic pathways.

## Spatially controlled myoblast/endothelial co-differentiation

We then asked if we could engineer a tissue construct in which multiple distinct cell fates are arranged in user-specified geometries, starting from a uniform, genetically identical cell population. To do so, we first engineered a "dual-lineage" cell line with two synNotch pathways: an anti-GFP/tTA synNotch receptor that activates myoD-miRFP and an anti-mCherry/Gal4 synNotch that activates ETV2-BFP (Fig. 7A(ii)). To test the functionality and orthogonality of these pathways, we cultured these dual-lineage cells on surfaces with a uniform coating of GFP or

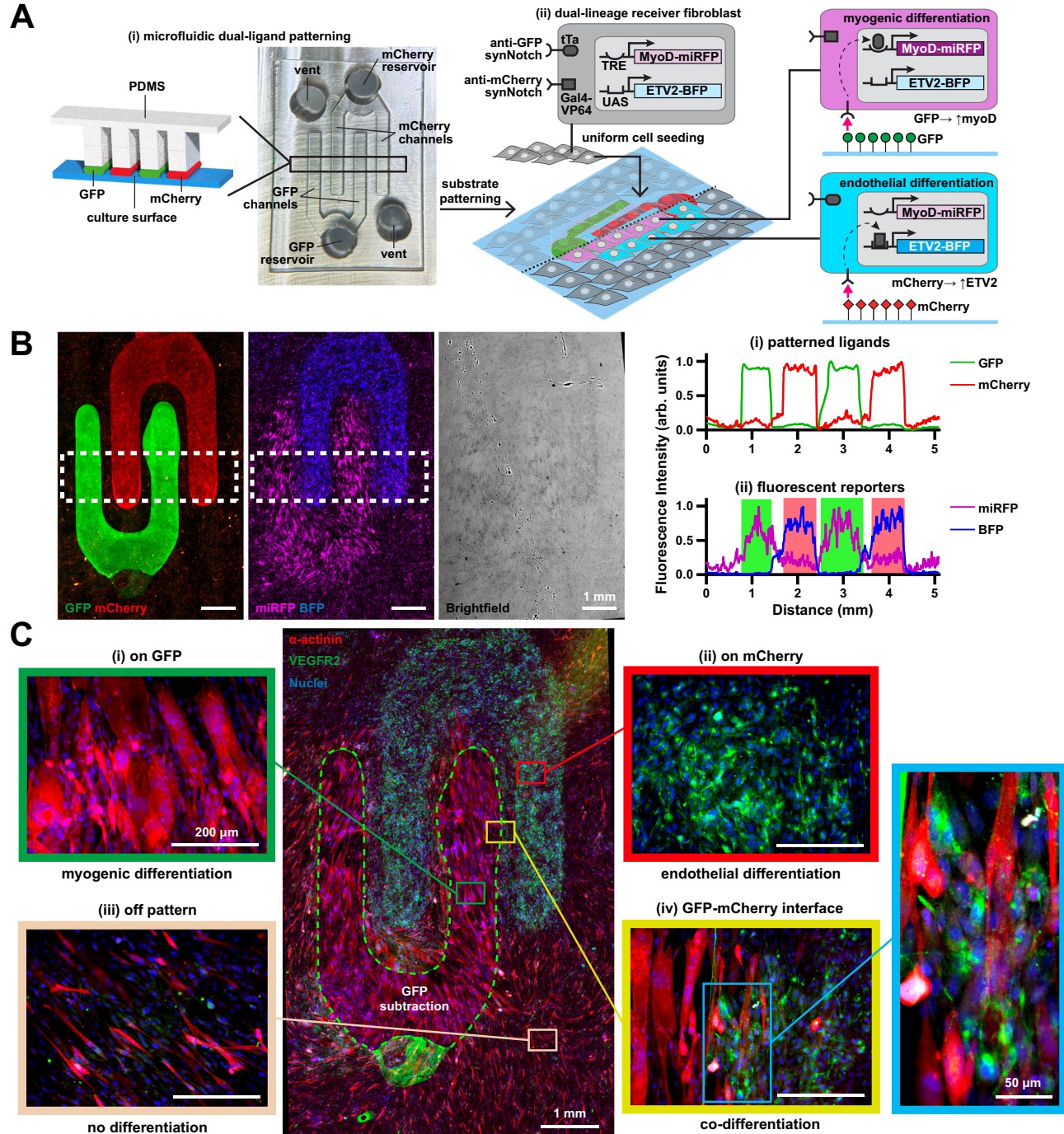

**Fig. 7 | Spatially controlled co-differentiation of dual-lineage receiver fibroblasts into myogenic and endothelial lineages via synNotch by microfluidically-patterned GFP and mCherry ligands. A** (i) Schematic of microfluidic patterning technique used to fabricate substrates with alternating rows of GFP and mCherry and photograph of corresponding microfluidic device. Channels are 500 μm wide. (ii) Schematic of dual-lineage receiver fibroblasts with anti-GFP/tTA synNotch that activates MyoD and miRFP and orthogonal anti-mCherry/Gal4-VP64 synNotch that activates ETV2 and BFP, cultured on corresponding substrates patterned with GFP and mCherry. **B** Left: Fluorescence microscopy images of substrate microfluidically-patterned with GFP and mCherry and corresponding expression of miRFP and BFP reporter genes by dual-lineage receiver fibroblasts cultured for three days on the substrate. Brightfield image of cells also shown. Scale bars, 1 mm. Dotted white rectangles represent regions of interest for profile plots on the right. Right: profile plot of (i) GFP and mCherry ligands and (ii) miRFP and BFP reporter genes. Green and red bars indicate regions containing GFP and mCherry, respectively. **C** Center: Fluorescence microscopy image of dual-lineage receiver cells cultured on substrates microfluidically-patterned with GFP and mCherry for 3 days and immunostained for α-actinin (red) and VEGFR2 (green). Dotted green lines indicate region where fluorescence from GFP ligand was subtracted due to high saturation to enable visualization of VEGFR2 fluorescence. See Fig. S9E. Scale bar, 1 mm. Representative regions patterned with GFP only (i, green square), mCherry only (ii, red square), neither GFP nor mCherry (iii, beige square), and GFP-mCherry interface (iv, yellow square), are shown at higher magnification. Scale bars, 200 μm. Further magnified image of GFP-mCherry interface is shown in blue square. Scale bar, 50 μm. Images representative of results observed in 4 repeated experiments with similar results. Source data are provided as a Source Data file.

mCherry for three days and then stained them for markers of differentiation. As shown in Figs. S8A–C and S9A, cells transdifferentiated to α-actinin-positive muscle precursor cells or VEGFR2-positive endothelial precursor cells on GFP or mCherry, respectively. As a curiosity, we evaluated the effects of culturing cells on both ligands, which would potentially induce overexpression of both myoD and ETV2 in the same cells. In this case, it seemed that transdifferentiation to both pathways was impaired, as these cells did not differentiate towards skeletal muscle nor express endothelial cell markers (Fig. S9A). To prototype simple spatial activation, we used a micropipette to deposit droplets of GFP and mCherry in opposing corners of a culture surface (Fig. S9B–D). Dual-lineage cells cultured on this surface activated the fluorescent reporters with expected spatial localization and displayed multinucleation exclusively in the GFP-coated region, indicating feasibility for spatial activation of differentiation.

Our next goal was to pattern multiple synNotch ligands onto a surface simultaneously and with spatial control. To do so, we adapted approaches for controlling the distribution of multiple streams of liquids with an open capillary microfluidic device[49]. Briefly, the intended fluid paths are created as shallow channels that are laterally open and adjacent to deep channels. Fluids preferentially travel along the shallow channels instead of the deep channels because of greater surface tension in shallow channels. We used this concept to design a microfluidic device for delivering solutions of GFP and mCherry by interdigitating 500-μm-wide rows (Fig. 7A(i)) and fabricated it by casting PDMS on 3-D printed inverse templates. Air vents and GFP and mCherry reservoirs were punched into the PDMS and the device was attached to a culture surface and loaded with GFP and mCherry solutions. After overnight incubation, the PDMS device was removed and the remaining solutions were briefly air-dried, leaving behind interdigitating rows of GFP and mCherry adsorbed on the surface (Fig. 7B). When dual-lineage fibroblasts were cultured on these surfaces, cells adhered uniformly to the entire surface after one day of culture.

After three days of culture, fibroblasts transdifferentiated to myoblasts or endothelial cells in a pattern corresponding to the intended pattern of ligands (Figs. 7B, C and S9E–F). As highlighted in Fig. 7C, α-actinin-positive muscle precursor cells were confined to the GFP rows, VEGFR2-positive endothelial precursor cells were confined to the mCherry rows, and intermixing of these two cell types was observed at the interface between GFP and mCherry. Cells on the unpatterned regions remained fibroblasts, although some of them showed some positivity for α-actinin, indicating some level of basal activation of the synNotch myoD pathway, as observed above.

To further evaluate the extent of dual-lineage transdifferentiation, we performed single-nuclei RNA sequencing on the dual-lineage cell line after three days of culture on substrates with no ligand, GFP-only rows, mCherry-only rows, and interdigitating GFP-mCherry rows (Fig. 8A), patterned using the technique shown in Fig. 7A. Based on gene expression profiles from all four conditions, T-Distributed Stochastic Neighbor Embedding (tSNE) plot analysis identified twelve cell clusters (Fig. S10A). We analyzed the clusters for signature genes and performed pathway analysis with DAVID[50,51] (Table S1) to assign each cluster to a putative cell type identity, resulting in seven fibroblast clusters, four muscle-like clusters, and one endothelial-like cluster. The presence of a large fibroblast cluster, even in the induced conditions, is expected since, as shown in the schematic in Fig. 8A, the area patterned by ligands occupies only approximately half of the culture surface. We also noticed the presence of some cells in muscle-like clusters, even in conditions without ligands, which is consistent with high basal activation of the synNotch myoD pathway. However, as shown in Fig. 8B, cells were induced towards the myogenic lineage on GFP-only patterns and more cells were induced towards the endothelial lineage on mCherry-only patterns, as expected. On the dual GFP-mCherry pattern, both myogenic and endothelial clusters were detected (Fig. 8A, B). Selected marker gene analysis (Fig. 8C) showed that fibroblast marker genes were overall down-regulated on patterns

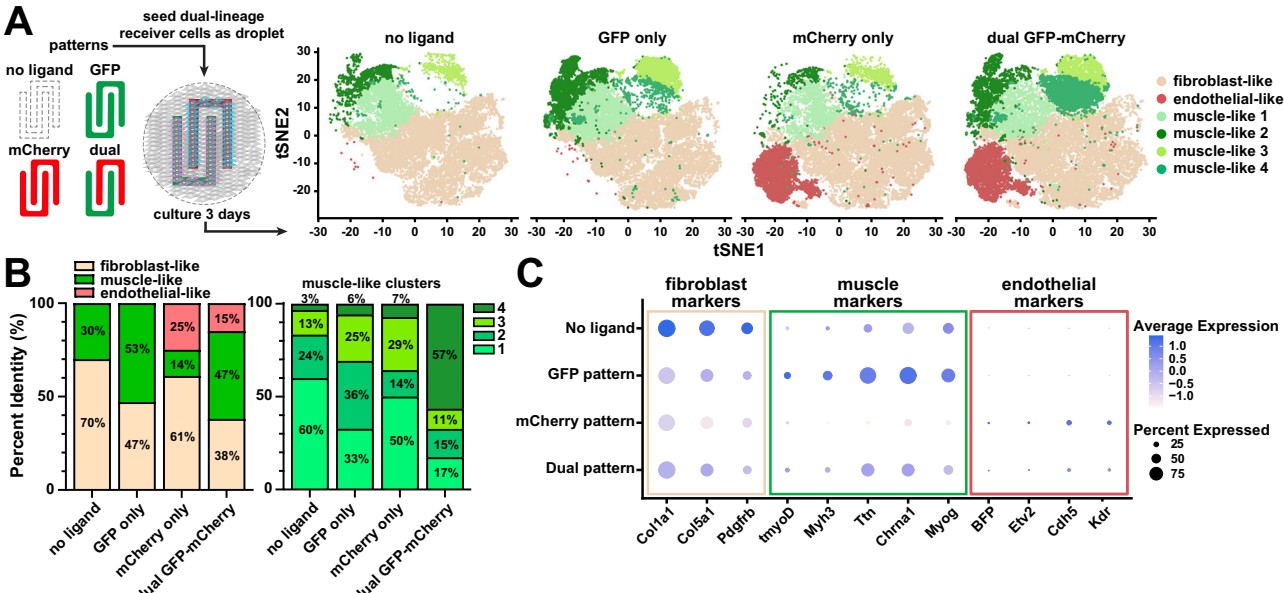

**Fig. 8 | Single-nuclei RNA sequencing of dual-lineage fibroblasts on patterned GFP and mCherry substrates. A** Left: Schematic showing the different ligand patterns used in single-nuclei sequencing experiments. Right: T-Distributed Stochastic Neighbor Embedding plot results of dual-lineage fibroblasts cultured on the four different patterning conditions. The fibroblast-like cluster contains seven individual clusters, shown here as one beige color (See Fig. S10A and Table S1 for more details). *n* = 2 biological replicates per condition. **B** Left: Percent of Fibroblast-like, muscle-like, and endothelial-like cells across the four patterning conditions.

Right: Percent ratio of muscle-like clusters make up of the total muscle-like cells in each patterning condition. (See Table S2 for more information on the muscle-like clusters). *n* = 2 biological replicates per condition. **C** Plot showing average expression and percent expression of selected fibroblast, muscle, and endothelial markers in all clusters across different patterning conditions. *n* = 2 biological replicates per condition. Source data available in the NIH GEO database under accession code GSE269404.

with GFP and/or mCherry, as expected. Correspondingly, muscle-specific genes and endothelial-specific genes were overexpressed on patterns with GFP and/or mCherry, respectively. With this analysis, we also detected the expected expression of the transgenes (transgenic myoD and BFP). Interestingly, on the dual pattern, we observed more cells in the muscle-like 4 cluster compared to the other three patterns, indicating that this cell identity may be unique to co-differentiation. The four muscle clusters all express similar muscle marker genes, but at different relative levels (Fig. S10B). Pathway analysis of differentially expressed genes revealed the four muscle clusters differ in pathways related to cell cycle, ribosome, and differentiation (Table S2), suggesting that these four clusters may represent similar cells at slightly different phases of the cell cycle or stages of differentiation. Alternatively, co-differentiation may have unique impacts on cell phenotype, but additional replicates and/or longer culture times are needed to reach a more clear conclusion. Thus, in summary, by activating synNotch receptors with microfabricated biomaterials, we induced a single population of fibroblasts to differentiate into tissue with three distinct cell populations (skeletal muscle precursors, endothelial precursors, and fibroblasts) patterned in user-defined microscale geometries with three days of culture. Of note, these tissues were maintained in standard cell culture media, without the need for soluble differentiation factors or biophysical stimulation to drive cell fates.

## Discussion

synNotch was originally developed in cell lines and primary T-cells for applications in cell therapy[52–56]. More recently, it has been implemented in different cell types, including a transgenic mouse where synNotch is used for contact-dependent labeling of cells including endothelial cells, hepatocytes, fibroblasts, pericytes, in vivo tissues[57]; and embryonic stem cells[58] where it was used for controlling differentiation. In this study, we engineered several material-to-cell signaling pathways to spatially activate user-defined genetic programs in multicellular systems. We achieved this by engineering cells with synNotch receptors to define cellular inputs and outputs while concurrently engineering materials to present synthetic ligands with different ranges of spatial control. The variety of materials for synthetic ligand presentation yields powerful and highly flexible tools for activating material-to-cell pathways. Due to the functional modularity of synNotch receptors, material-activated pathways can theoretically be used to drive any number of transcriptional programs or differentiation pathways. These generalizable technologies are an approach for dictating spatial patterning of gene expression in multicellular constructs, without the need for soluble differentiation factors.

Because of the highly powerful level of transcriptional control over cell behaviors in natural systems, many efforts in synthetic biology have focused on engineering sophisticated transcriptional circuits[59,60]. In the area of stem-cell and cell differentiation, genetic overexpression of master transcription factors has demonstrated robust control over cell differentiation[61,62]. Initially, the effect of only a handful of master transcription factors on cell differentiation was known. However, more recently, approaches that collect entire organism transcription-factor libraries have become available, making it feasible to induce multiple differentiation pathways with technologies such as synNotch[63,64]. The capacity to induce master transcription factors with user-defined spatial control has inspired several recent advances, such as engineering cells with light-activatable signaling pathways to gain spatiotemporal control over cell behaviors with light[65,66]. For example, myoD overexpression has been induced by a genetically integrated optogenetic switch that can be activated spatially[67]. Although optogenetic approaches have the potential for powerful spatiotemporal control over cell behaviors, and have recently been shown to be multimerized to up to three orthogonal pairs[68], optogenetic technologies require sophisticated light manipulation devices, which can be difficult to scale and have limited penetration into 3-D tissues, and have not yet demonstrated robust multi-cell fate control.

With our previous development of synNotch, we generated a way to activate user-defined genetic programs via user-defined ligands presented by neighboring cells. Here, we advanced this technology to a new level by activating synNotch via multiple materials commonly used for tissue engineering. Importantly, we showed that this approach can be used to define spatial patterns of not only gene expression, but also differentiation. To present synthetic ligands, we modified several different types of materials, each with tradeoffs. By engineering cells to secrete fusions of synthetic ligands and the natural ECM protein fibronectin, ligands are presented in a natural ECM network comprising a diversity of endogenous macromolecules, which may enhance receiver cell adhesion and survival. However, spatial control is very coarse, as spatial ECM deposition by cells is not fully understood or controllable. Hydrogels are the most common class of materials for tissue engineering due to their high water content and multiple tunable properties, including stiffness, porosity, and composition[69]. Thus, we developed versatile and modular methods for presenting synthetic ligands via hydrogels, by using relatively simple enzymatic reactions or click chemistry reactions to conjugate GFP or mCherry on the surface of gelatin or fibrinogen in 2-D cultures or within bulk hydrogels for 3-D cultures. One limitation here is that spatial patterning of ligands in hydrogels was limited to manual pipetting, which is relatively coarse. However, many other modalities exist to conjugate and/or release proteins from hydrogels with spatial control[70], which can be integrated in future iterations.

Bulk RNA sequencing and principal component analysis of single-lineage receiver cell lines on materials with single ligands validated that cells were transdifferentiating towards the intended myogenic or endothelial lineages. However, we also observed cells with synNotch activating MyoD clustered further from the unmodified parental fibroblast cell line compared to cells with synNotch activating ETV2, without any ligand. This is likely due to the leakiness of the MyoD transgene and/or the strength of the transcription factor itself relative to ETV2. Different transcription factors likely have different activation amplitudes and dynamics and it would be crucial to identify appropriate signal-to-noise ratios for each specific application by, for example, generating synNotch receiver cells with different amounts of receptor and target gene constructs and assessing experimentally which combination works more efficiently for the transgene of interest. Future advancements in computational modeling and design can likely also help screen these combinations more efficiently.

In terms of heterogeneity, we observed bimodal and therefore incomplete activation of synNotch by ligands presented by materials, similar to other studies that have presented synNotch ligands from cells or other materials. Across all materials we tested, we found that synNotch activation reached a plateau in response to increasing ligand concentration, beyond which synNotch activation did not increase. Thus, we likely reached the saturation point of ligand presentation by the material and synNotch signaling itself seems to be the main factor limiting activation. We correspondingly observed an imperfect differentiation efficiency, which is likely a compounded effect of the heterogeneity of synNotch activation and the known heterogeneity of transcription-factor-mediated differentiation, especially at the early time points that we investigated in this study. These are major limitations of synNotch but will continue to improve as the technology evolves.

In terms of dynamics, we found that different ligand-presenting materials yielded different temporal patterns of synNotch activation. For example, synNotch activation peaked at three days and then subsided when ligands were microcontact-printed on PDMS, whereas activation was more sustained when synNotch was activated by ligands conjugated to 3-D hydrogels. This could be caused by differences in the conjugation of the ligands to the materials, such as the strength of

the material-ligand bond or ligand orientation, and/or differences in ligand-receptor engagement and the activation of the synNotch receptor itself. The mechanism of transduction by synNotch receptors is thought to proceed similarly to endogenous Notch receptors: there, in the core regulatory region of endogenous Notch receptors, a pulling force is generated upon ligand binding, which exposes a protease cleavage site for a protease that is constitutively active in the membrane; this cleavage then liberates the intracellular domain which is a transcription co-activator[26,71]. The mechanisms of activation of Notch and synNotch receptors via cell-presented ligands have been compared, individuating possible mechanisms of activation that distinguish different synthetic and natural receptor constructs[72]. Thus, the mechanism of activation of synNotch by material-presented ligands may also differ from cell-presented ligands and may differ for different materials with various chemical and mechanical properties. Increased mechanistic understanding of synNotch activation by materials could yield increased capacity for spatial, and perhaps temporal, control of gene expression.

Another interesting result from our study is the impact of both ligands on dual-receiver cells. For cells with two synNotch pathways that activate fluorescent reporters, both reporters were expressed in cells cultured on both ligands. However, for cells with two synNotch pathways that active myoD or ETV2 cultured on both ligands, both myogenic and endothelial differentiation programs were stunted (see Fig. S9A). To achieve dual differentiation in the presence of two ligands, lineage bifurcation moduli (such as lateral-inhibition) or cross-inhibition to prevent the opposing lineage could lead to a salt-and-pepper or checkerboard pattern of differentiation in regions with both ligands. These types of approaches could also be combined with the MATRIX system to provide ligands at specific time points and achieve more advanced spatial and temporal control over differentiation, resulting in more complex tissue patterns.

To achieve greater spatial control, we microfabricated PDMS stamps and microfluidic devices to pattern synthetic ligands onto 2-D surfaces. By fabricating these components on 3-D printed templates instead of classical photolithography-based wafers, we achieved a wider range of pattern designs and more rapid prototyping capabilities, with the tradeoff that spatial resolution was confined to 100 μm or above. To pattern synthetic ligands at sub-cellular spatial resolution, photolithography would still be required. The two PDMS-based patterning technologies that we used also have tradeoffs. Microcontact printing can generate essentially any geometrical pattern (including isolated islands) but cannot precisely register multiple ligands since each stamp must be positioned manually. Conversely, registering the placement of multiple ligands is possible with a microfluidic device, but pattern geometries are limited to continuous channels connected to a reservoir. Thus, these constraints must be considered when choosing a patterning modality. Together with other approaches, such as MATRIX[27], these new approaches expand the library of engineered biomaterials that activate synNotch.

Our most sophisticated tissue construct comprised interdigitating rows of skeletal muscle and endothelial cells, with some intermingling of the cells at the interface. Importantly, the skeletal muscle cells and endothelial cells were co-transdifferentiated from a single population of fibroblasts. This approach is in contrast to conventional tissue engineering techniques, which usually differentiate individual cell types in isolation and then combine them. Our approach may better mimic natural tissue morphogenesis, where multiple cell fates emerge simultaneously from a uniform cell population. Studies have also shown that supporting cells, such as endothelial cells, improve the maturation of human induced pluripotent stem-cell-derived cardiomyocytes[73,74], and that co-differentiation of different lineages concurrently more closely recapitulate the conditions occurring during embryonic development[75]. An interesting hypothesis to explore with our technology is whether co-differentiation of supporting cells

(e.g., endothelial cells) adjacent to parenchymal cells (e.g., muscle cells) has additional benefits for phenotypic maturity. For example, our single-nuclei sequencing revealed one muscle-like cluster that was overrepresented on the dual-ligand pattern compared to the GFP-only pattern. There are many potential explanations for this, such as: (i) these muscle-like cells were uniquely influenced by the presence of the co-differentiating endothelial cells; (ii) these muscle-like cells were located at the GFP-mCherry boundary and thus were activated predominantly by GFP but also by mCherry to a lower extent; and/or (iii) these muscle-like cells were coincidentally captured at a unique stage of cell cycle or differentiation but are otherwise similar to the other muscle-like clusters. To tease apart these different possibilities, additional replicates and longer time points will be needed. Furthermore, the differentiation protocols we used here were very simplified; i.e., only three days of differentiation in a basal medium with no additional soluble differentiation factors. Initial cell state changes are expected in this timeframe, based on previous studies with induced transcription factors, but more complete lineage conversion to mature cell types will require more time for differentiation and potentially supplementation with soluble differentiation factors for some cell types.

Patterning multi-lineage tissues with spatial control has also been achieved with multi-material extrusion bioprinting. In a recent example, human stem cells were engineered with doxycycline-inducible transcription factors for endothelial or neural cell fates[76]. Wildtype or engineered cells were embedded in individual bioinks, merged into a tri-layer filament, and extruded through a nozzle in user-defined patterns. Doxycycline and differentiation media were then added to induce the co-differentiation of neural stem cells, endothelial cells, and neurons. Although this is a powerful approach for multi-lineage tissue engineering, it does have its own limitations, such as the reliance on diffusion-limited soluble differentiation factors, restrictions on spatial resolution imposed by the nozzle, and a somewhat limited library of printable materials[77]. However, we envision many synergistic opportunities for bioprinting and synNotch technologies to be used together by, for example, bioprinting hydrogels that are functionalized with synNotch ligands, such as the gelMA and fibrinogen that we synthesized in this paper. Optogenetic technologies[68], as mentioned above, can also be integrated to add more temporal control of cell phenotype. Overall, the ongoing integration of synthetic biology, biomaterials, and microfabrication technologies will further advance the capabilities for tissue engineering[78].

Beyond bioprinting, we anticipate that our approach for activating synthetic pathways for transdifferentiation by a material can be combined with other complementary technologies for deriving complex in vitro tissues, such as organoids[79]. To generate organoids, stem cells are exposed to natural ligands that orchestrate their self-organization into complex cellular arrangements. However, although cellular complexity at the microscale in organoids is remarkably similar to endogenous organs, users lack geometric control over the arrangement of cells at higher levels, leading to tissue constructs that are largely heterogeneous and poorly reproducible with unnatural architectural features. Combining synthetic biology and organoids is a recognized frontier of the field[12,80–83] and synNotch-mediated spatial patterning technologies, such as those presented here, could represent a step in the direction of ultimate user control of cell behaviors across multiple spatial scales for engineering in vitro multicellular systems.

## Methods
### Genetic constructs design
Fibronectin-GFP plasmids were generated from the PiggyBac backbone and FN-YPET (Addgene #65421). GFP and mCherry-responsive synNotch construction: pHR_SFFV_myc-LaG17_synNotch_TetRVP64 (Addgene plasmid# 79128) and pHR_EF1a_flag-LaM4_synNotch_Gal4-VP64, built from pHR_EF1a_flag-LaM4_synNotch_TetRVP64 (Addgene

plasmid#162237) and HR_pGK_LaG17_synNotch_Gal4VP64 (Addgene plasmid# 79127). The response-element plasmids pHR_TRE_MyoD-P2A-mCherry, pHR_TRE_MyoD-P2A-miRFP703_PGK_PuromycinR, and pHR_UAS_ETV2-P2A-tBFP_PGK_HygromycinR (with and without transcription factor) were generated from pHR_TRE, pHR_5x Gal4 UAS (Addgene plasmid# 79119), mouse MyoD (NP_034996.2), and mouse ETV2 (NP_031985.2, Twist Bioscience). All constructs were cloned via In-Fusion HD Cloning (Takara Bio, 102518).

Oligonucleotide primer sequences are provided in the supplemental information file.

## Lentivirus production
Lentivirus was produced by cotransfecting pHR cloned plasmids with vectors encoding packaging proteins (psPAX2, pVSVG) using Lipofectamine LTX (Thermo Fisher A12621) into 70–80% confluent HEK-293T cells within 6-well plates. Viral supernatants were collected 2–3 days after transfection, sterile filtered with 0.45 μm PES (Genesee Scientific), and used directly or 10× concentrated using LentiX Concentrator (Takara Bio, 631232) following the manufacturer's instructions prior to adding to cell lines.

## Cell culture
L929 mouse fibroblast cells (ATCC# CCL-1), HEK293 cells (Takara 632180), C3H/10T1/2 Clone 8 (ATCC# CCL-226), and NIH/3T3 (ATCC# CRL-1658) were cultured in DMEM (Thermo Fisher) supplemented with 10% Fetal Bovine Serum (Thermo Fisher) and 100 U/mL penicillin/streptomycin (Thermo Fisher). Cultures were maintained in a 37 °C incubator with 5% $CO_2$ and relative humidity (VWR).

## Cell line engineering
For the generation of 3T3 fibroblast expression FN-GFP, 20,000 3T3 cells were seeded in a 12-well plate. The following day cells were transfected with 1 μg FN-eGFP PiggyBac plasmid using 2.5 μL Lipofectamine LTX with 1 μL Plus reagent diluted in 100 μL OptiMEM. Transfected cells were selected using 2 μg/mL Puromycin. Additionally, the established line was transduced with lentivirus encoding the expression of constitutive H2B-miRFP703.

For viral transduction, 20–50 μL concentrated (or equivalent non-concentrated) viral supernatant(s) were added to 5–10 × 10$^4$ suspended cells supplemented with 10 μg/mL polybrene (Sigma), then transferred into a 12-well plate for 2–3 days before changing to fresh media. Following transduction, all applicable cell lines were selected using Puromycin (L929 – 10 μg/mL, C3H – 1 μg/mL, NIH3T3 – 2 μg/mL, Thermo Fisher) and Hygromycin B (L929, C3H – 400 μg/mL, Med-Chem Express) for the expression of transgenes. Cells were sorted for the coexpression of each component via fluorescence-activated cell sorting on a FACS ARIA II (Beckton-Dickinson) by staining with appropriate fluorescently tagged anti-Myc and anti-Flag antibodies for 30 min at 4 °C (Cell Signaling Technologies) or expression of the transgenes. A bulk-sorted polyclonal population of engineered cells was used for experiments unless otherwise noted. For single-cell clonal populations, single cells were sorted individually into 96-well plates from selected and stained populations using a FACS ARIA II.

## GFP and mCherry production
GFP, mCherry, and GFP-LACE (pET28-His6-GFP-C-LACE, gift from Jeffrey Bode Addgene plasmid # 133913) were purified as an N-terminal hexahistidine fusion protein. To express GFP, BL21(T1R) E. coli cells were grown to an optical density of 0.5 from an overnight-grown glycerol stock, chilled to 25 °C, induced with 1 mM IPTG (Sigma, I6758), and allowed to express for 5 h. To express mCherry, BL21-AI E.coli (Thermo Fisher, C607003) were transformed with mCherry-pBAD (gift from Michael Davidson & Nathan Shaner & Roger Tsien, Addgene plasmid # 54630), grown to an optical density of 0.6 from an overnight-grown glycerol stock, induced with 0.04% w/v L-Arabinose

(Sigma, 10839), and allowed to express for 5 h, based on previous studies[84]. The proteins were purified by NEBExpress Ni Spin Columns (New England Biolabs, S1427S) following the manufacturer's instructions, dialyzed against 1× PBS overnight at 4 °C, sterile filtered, and frozen at −80 °C until use.

## Microparticle-conjugation and activation
Carboxylated magnetic polystyrene microparticles (Magsphere, MCA5UM) were first washed in 0.1 M MES (pH 5.8), activated with 250 mM 1-Ethyl-3-(3-dimethylaminopropyl)carbodiimide (EDC, Sigma)/N-Hydroxysuccinimide (NHS, Sigma) in MES for 15 min, then washed two times with PBS (pH 7.4). Particles were then incubated in varying concentrations of GFP in PBS (0–1000 μg/mL) overnight at 4 °C with inversion mixing. All washing steps used EasySep Magnet (Stemcell Technologies) or centrifugation (8000 × g for 5 min) to change solutions. Particle concentration was determined using a hemocytometer and directly loaded into cell suspensions (10 particles per cell) prior to seeding L929 anti-GFP synNotch receiver cells 25,000 cells/cm$^2$ directly into wells or gelatin-coated coverslips. 24 h after seeding, samples were imaged directly using Zeiss Axio Observer Z1 or stained with NucBlue and fixed with 4% PFA for 10 min. Individual cell mCherry intensity was quantified on ImageJ using nuclear segmentation from 5 images per sample. Gaussian blur, thresholding, watershed, and analyze particle functions were applied to the nuclei and mCherry channel and to create individual selections for total cells and activated cells, respectively. The mCherry mask was applied to the corresponding mCherry image to measure the average fluorescence intensity within each activated cell. Percent activation was determined by the number of mCherry-positive cells divided by the total number of nuclei. Activated mCherry intensity was calculated by averaging the mCherry intensity for cells above the defined threshold.

## Fibronectin-GFP activation
For local activation, L929 anti-GFP synNotch receiver cells were seeded with 3T3 cells expressing FN-eGFP and nuclear-localized miRFP703 in a ratio of 50:1 to a total of 5 × 10$^4$ cells per well on an 8-well slide (Ibidi). After 3 days, cells were imaged on a Zeiss LSM780. For the titration experiments, cells were seeded in 8-well ibidi slides coated with 0.1% gelatin to a total of 3 × 10$^4$ parental 3T3 cells and FN-GFP-sender cells in the following ratios 1:0, 50:1, 5:1, 2:1, 1:1 and 0:1 (Parental: FN-GFP). Following 8–10 days of culture, decellularization was performed following previously established methods[42]. Briefly, cell-laden ECM was washed with PBS, wash buffers, and a lysis buffer containing NP-40 for up to two hours to remove cellular debris. Removal of nuclear debris was confirmed by Hoechst staining prior to decellularization, which was used to monitor decellularization quality during the lysis phase of the protocol. Decellularized ECM was used immediately or stored at 4 °C until use. 5–10 × 10$^4$ L929 anti-GFP-synNotch receiver cells were seeded onto the decellularized matrices. After 2 days cells were imaged on a Zeiss LSM780 or Keyence BZ-X and on the same day analyzed by FACS on a Thermo Fisher Attune. For other co-culture experiments, L929 anti-GFP (or anti-mCherry) synNotch receiver cells were seeded at a 1:1 ratio with FN-GFP or FN-mCherry senders. Activation of engineered cells was imaged using Keyence BZ-X or fixed, stained for fibronectin (primary O/N 4 °C, secondary 1 h RT), and imaged on a Zeiss LSM780.

## Gelatin hydrogel surface conjugation
Gelatin hydrogels were fabricated as previously described[32,34]. Briefly, a 30 W Epilog Mini 24 laser engraver (100% speed, 25% power, 2500 Hz) was used to cut a 150-mm polystyrene dish into 260-mm$^2$ hexagons. Each hexagon was masked with tape, and an inner circle was cut (18% speed, 6% power, 2500 Hz) and removed, exposing a polystyrene surface which was then treated with plasma (Harrick Plasma) for 10 min to improve gelatin adherence to polystyrene. Equal volumes of

a 20% porcine gelatin solution (Sigma) and 8% MTG (Ajinomoto) solution were mixed and 200 µl were added to each coverslip. Flat or 10 × 10 µm micromolded PDMS stamps were immediately applied to shape surface topography. After an overnight incubation to solidify, the hydrogels were rehydrated in water, and the stamp was removed. Coverslips were stored in PBS at 4 °C until cell seeding.

PDMS stamps with 10 × 10 µm grooves of 2 µm height were fabricated with standard photolithography and soft lithography techniques[43]. Flat PDMS was used as a control substrate with no topography. A 1:1 ratio of GFP (500 µg/ml, 200 µg/ml, and 20 µg/ml) and MTG (8% w/v) solution were added to a parafilm surface, and the gelatin coverslip was inverted onto the GFP-MTG droplet for 10 min[34]. The coverslips were then incubated for 1 h at 37 °C. Following incubation, the coverslips were washed 3 times with warm PBS.

L929 anti-GFP synNotch receiver cells were seeded at a density of 350,000 cells per coverslip and cultured for 72 h. To detach the cells from the gelatin for flow cytometry analysis, the gelatin hydrogels were minced with a sterile X-acto knife and incubated in a 4 mg/ml collagenase IV solution for 45 min at 37 °C. Digested gelatin was then filtered with a 40 µm cell strainer.

C3H anti-GFP synNotch MyoD expressing cells were seeded at a density of 500,000 cells per coverslip, and cultured for 4 days. Coverslips were washed three times with warm PBS, fixed with ice-cold methanol, and immunostained with mouse α-actinin primary antibody (Sigma, A7811) at 1:200 dilution for two hours. Coverslips were then stained with the secondary antibody goat anti-mouse conjugated to Alexa Fluor 546 and 4′,6-diamidino-2-phenylindole (DAPI) at 1:200 dilutions for 2 h. ProLong gold-antifade mountant (Thermo Fisher) was used to mount cells on glass coverslips.

Myotube count was performed in ImageJ through size and intensity thresholding of the α-actinin signal. The total number of cells per image was calculated by size and intensity thresholding of the DAPI signal. The myogenic index was determined by dividing the number of co-localized nuclei within the α-actinin signal by the total number of nuclei in the field of view. The Orientation Order Parameter of both the myotubes and nuclei was determined by first analyzing images of the α-actinin and DAPI signal, respectively, using the OrientationJ Distribution plugin in ImageJ[85]. This plugin was used to generate a histogram with the number of pixels locally oriented along every angle at 0.5° increments. This histogram was then analyzed using MATLAB code to calculate the Orientation Order Parameter[45], which ranges from 0 for completely randomized systems to 1 for perfectly aligned systems.

## 3-D ligand conjugation

GelMA Synthesis: Porcine Gelatin (175G Bloom, Sigma G2625) was dissolved at $10 \times g$ in 100 mL in 0.25 M carbonate-bicarbonate buffer (pH 9) at 50 °C under argon. 0.4 mL methacrylic anhydride (Sigma 276685) was added dropwise with stirring (500 rpm) and reacted for 3 h at 50 °C[86]. The reaction was then cooled to 40 °C, adjusted with 6 M HCl to pH 7.4, transferred to 12–14 kDa cutoff dialysis tubing (Fisher Scientific), dialyzed for 3 days at 40 °C against 4 L deionized water (changed twice daily), and lyophilized. GelMA was stored at −80 °C until use. The degree of methacrylation was calculated using $^1$H-NMR compared to unmodified gelatin.

GelMA was dissolved at 1% w/v in PBS at 37 °C. Methyltetrazine (mTz)-PEG5-NHS Ester (Click Chemistry Tools, 1069-100) was first dissolved at 8.8 mM in DMSO and added to GelMA solution dropwise with stirring to 0.88 mM final concentration. The mixture was reacted at 37 °C overnight, transferred to 12–14 kDa cutoff dialysis tubing, and dialyzed for 3 days at 40 °C against 4 L deionized water (changed twice daily), and lyophilized. The substitution was verified using $^1$H-NMR compared to unmodified GelMA and used to estimate the percent substitution of methacrylate and methyltetrazine. Ligands (GFP and mCherry) were modified using trans-Cyclooctene (TCO)-PEG4-NHS

Ester (Click Chemistry Tools, A137-2) to generate GFP-TCO and mCherry-TCO. TCO-PEG4-NHS Ester was dissolved at 10 mM in DMSO, added at a 20-molar excess to GFP and mCherry in PBS (1–3 mg/mL), and reacted for 1 h at room temperature in Eppendorf tubes with orbital rocking. Following, the mixture was purified using Zeba Spin Desalting Columns (7k MWCO, Thermo Fisher) following manufacturer instructions, sterile filtered, and aliquoted to 1 mg/mL and stored at −80 °C. Reactivity was verified by reacting GelMA-mTz at 1% w/v and GFP-TCO at 100 µg/mL for one hour at 37 °C, running on 4–20% SDS-PAGE gel (Bio-Rad) and Coomassie blue staining (Invitrogen).

## Cell encapsulation and activation

GelMA-mTz at 1% w/v was reacted with GFP-TCO at 50–100 µg/mL for one hour at 37 °C. Following, 18% w/v GelMA solution and freshly prepared 25 mg/mL Lithium phenyl-2,4,6-trimethylbenzoylphosphinate (LAP, Sigma 900889) was added to a final concentration of 5–10% w/v GelMA, 0-100 µg/mL GFP-TCO, and 0.25% LAP. Engineered L929 or C3H cell lines were resuspended in this solution at 5–10 × 10⁶ cells/mL. An array of 20 µL droplets of cell-laden hydrogel solutions were pipetted between two 25 × 75 mm glass slides that were thrice treated with GelSlick (Lonza) separated by 3-D-printed 400 µm insert. Gels were crosslinked for 90–180 s at 25 mW/cm² using an Omnicure2000 and collimating adapter (Excelitas) and transferred to individual wells with DMEM complete. Following 30 min incubation at 37 °C, media was exchanged to fresh DMEM complete. Cell-laden hydrogels were cultured for up to 14 days with media changes every 2 days. In the hydrogel patterning experiment, cell-laden hydrogel solutions containing 0 or 50 µg/mL GFP were pipetted in proximity to each other to initiate limited contact immediately prior to crosslinking. This process led to a stable biphasic gel that was treated similarly as described above. Samples were imaged live with Zeiss Axio Observer Z1 or Zeiss LSM880 confocal microscope. Individual cell mCherry intensity was quantified on ImageJ using constitutive BFP-signal to segment individual cells. Gaussian blur, thresholding, watershed, and analyze particle functions were applied to the BFP channel to create individual selections for each cell. This mask was applied to the corresponding mCherry image to measure the average fluorescence intensity within each cell. Percent activation was determined by the number of BFP+ cells that had a mCherry intensity higher than a defined threshold. For spatial activation within 3-D, line plots were quantified using the "Plot Profile" feature on inverted microscope images in ImageJ and normalized across the entire time course. Evaluation of cell viability was performed using a Live/Dead Viability/Cytotoxicity Kit (Thermo Fisher, L3224) following the manufacturer's instructions. Briefly, cell-laden gels were incubated in a 2 µM calcein AM and 4 µM EthD-1 solution in PBS for 45 min prior to imaging on a Zeiss LSM880.

In the co-culture encapsulation experiment, GFP-sender fibroblasts and anti-GFP synNotch receiver cells were co-encapsulated within 5% GelMA hydrogels at a fixed total cell concentration of 20 × 10⁶/mL. The ratio of each cell type was varied. Hydrogels were imaged on an LSM880 and the percent mCherry expression was evaluated using image analysis, where the BFP signal was used to segment individual receiver cells.

Fibrinogen (Sigma, F8630) was dissolved at 5 mg/mL in warmed PBS. Methyltetrazine (mTz)-PEG5-NHS Ester (Click Chemistry Tools) was first dissolved at 20 mM in DMSO and added to the Fibrinogen solution dropwise to a 0.16 mM final concentration. The solution was incubated at room temperature with manual rocking every 10 min for one hour, transferred to 12–14 kDa cutoff dialysis tubing (Fisher Scientific), and dialyzed overnight at 4 °C against 12 L deionized water, and lyophilized. The resulting Fibrinogen-mTz was stored at −80 °C until use. For generation of Fibrinogen-mCherry and cell encapsulation, Fibrinogen-mTz was dissolved at 20 mg/mL in PBS and incubated

with 160 μg/mL mCherry-TCO in PBS for one hour at 37 °C. DMEM was added to bring to a final 10 mg/mL Fibrinogen + 100 μg/mL mCherry and used to resuspend cell pellet at $5 \times 10^6$ anti-mCherry synNotch receiver cells/mL. Immediately prior, 0.2 U Thrombin/mg Fibrinogen was added to the solution and samples were allowed to gel for 10 min at 37 °C. Following a 15 min wash with DMEM to remove unbound mCherry, samples were transferred to the incubator for culture. Following 24 h of culture, HSC Nuclear Mask Deep Red was added for one hour at 37 °C to visualize nuclei before imaging on Zeiss Axio Observer Z1.

### Microcontact printing

For all surfaces except Fig. S6B (see below), 18 mm glass coverslips were spin-coated with PDMS, treated with UV ozone for 3 min, and incubated in 10% APTES in ethanol for 2 h at 50 °C. Coverslips were then rinsed with water and incubated in 2% glutaraldehyde solution in ethanol at room temperature for 1 h[41]. Coverslips were then rinsed again and inverted onto 150 μL droplets of 50 μg/mL fibronectin in distilled water in a Petri dish, which was then sealed with Parafilm and incubated overnight at 4 °C. Cylindrical isotropic stamps were cut from a slab of PDMS using an 8 mm diameter biopsy punch. GFP at 0, 10, 50, 100, or 200 μg/mL was coated onto the isotropic stamps and left for 1.5 h at room temperature until the solution was dry. The stamps were then briefly dipped into sterile water, air-dried using compressed air, and inverted onto fibronectin-coated coverslips. C3H anti-GFP synNotch mCherry-expressing cells were seeded onto patterned coverslips at 650,000 cells per coverslip in a 12-well plate.

To generate micropatterns of GFP, Solidworks was used to design desired patterns (square arrays ranging from 100 μm to 1 mm, concentric circles, aligned rows, and letters), which were then printed into templates using a digital light processing (DLP) 3-D printer (CADworks3D). After 3-D printing, the templates were placed in 200-proof isopropyl alcohol overnight to ensure all uncured resin was removed. The templates were then UV-cured for 1 h (back side for 20 min, feature side for 40 min) to finalize the curing process. PDMS (Sylgard 187) was poured into the templates, desiccated for 30 min, and cured overnight in a 65 °C oven. PDMS stamps were then removed from the templates. Microcontact printing with these PDMS stamps was performed the same as with isotropic stamps at 100 μg/mL GFP. For perpendicular row patterns, one stamp with aligned rows was coated with GFP (200 μg/mL), and another similar stamp was coated with mCherry (200 μg/mL). These stamps were manually positioned sequentially in a perpendicular orientation. Coverslips were stored dry at 4 °C until use and incubated in DMEM + 10% FBS for a minimum of 1 h prior to cell seeding. Coverslips were then seeded with C3H anti-GFP synNotch mCherry-expressing cells, C3H anti-GFP synNotch MyoD expressing cells, or monoclonal dual-receiver cells at a concentration of 650,000 cells per coverslip in a 12-well plate.

For the surfaces in Fig. S6B, PDMS-coated coverslips were treated with UV ozone for 8 min, then microcontact-printed with a stamp coated with a mixture of 100 μg/mL GFP and 50 μg/mL FN. After patterning, coverslips were incubated in 2% Pluronic in distilled water for 15 min at room temperature and rinsed with PBS[45].

### Dual-ligand patterning with capillary microfluidic device

Solidworks was used to design a 4-row capillary fluidic device with two disconnected inlets. Shallow channels (100 μm distance from the substrate) were designed to guide protein solutions. These shallow channels were surrounded by 1 mm deep channels, intended as voids. The inverse design was 3-D printed using a DLP printer (CADWorks), which was then replica molded in PDMS. Inlets were created using 1.5 mm biopsy punches and air ventilation punches were created on two opposite sides ends of the device to allow optimal pressure for capillary fluid transfer. Prior to protein patterning, the feature side surface of the device was UV plasma treated for 7 s, creating a hydrophilic surface for capillary action. The device was then placed facedown onto tissue culture-treated Ibidi 2-wells. GFP (500 μg/mL) and mCherry (1000 μg/mL) were then pipetted into separate inlets and filled their respective shallow channels. The device was then placed into a petri dish and parafilm sealed before overnight incubation at 4 °C. The next day, the device was incubated without parafilm at room temperature for 15 min. The fluidic device was then carefully removed from the Ibidi wells to minimize liquid disruption. The Ibidi well was left at room temperature for 15 min for the protein solutions to dry. The Ibidi well was then UV-treated under the biosafety cabinet for 1 h to sterilize. DMEM with 10% FBS was pipetted into the wells and incubated for 1 h before cell seeding. Dual-lineage cells were seeded at $1.9 \times 10^5$/cm² and cultured for three days prior to fixation and staining for α-actinin and VEGFR2.

### Data quantification

We studied the spatial control dynamics over time by measuring Pearson's Coefficient on days 2, 5, and 10. This was done using the JACoP plugin in ImageJ by comparing the binary mask to the mCherry channel. The scrambled images were created in ImageJ using multiple binary masks and running a custom macro to scramble the pixels, provided as an ImageJ macro in "Code availability" section. The scrambled images were then compared to their respective mCherry images to quantify Pearson's Coefficient of the scramble condition. Line plots were quantified using the "Plot Profile" feature in ImageJ and normalized to individual images. The number of nuclei in the field of view of each image was counted, and the number of myotubes was counted using ImageJ. The myogenic index was calculated by dividing the number of nuclei within each myotube by the total number of nuclei. The GFP channel was used to determine which nuclei were on and off-pattern. For calculating the on-pattern myogenic index, all nuclei outside the GFP pattern were excluded. The sarcomeric α-actinin mask was overlaid on the on-pattern nuclei and used to calculate the myogenic index. Similarly, to quantify the off-pattern myogenic index, we excluded all nuclei located within the GFP patterns and used the same sarcomeric α-actinin mask to measure the myogenic index. For Coherency quantification, 200 and 500 μm rows and curves were thresholded using the same methods used to create the myotube mask, except instead of using the thresholded image to create a selection/mask, we quantified the thresholded myotube image itself. The OrientationJ plugin on ImageJ was used to quantify the coherency of all patterns. Since curved rows are not straight, we need to straighten them to get a fair quantification of how the myotubes align with the curves. A fragmented line was drawn manually following the GFP pattern of the curve and used to straighten the myotube threshold image before quantifying the coherency. All data was processed in GraphPad Prism 9 and validated with statistics.

### Plate-drying of ligand

For single-ligand activation, ligands (mCherry and GFP) were prepared at 100 μg/mL in sterile DI water and added at 15 μg/cm². For dual-ligand patterning, 8 μL droplets of ligand at 200 μg/mL in sterile DI water were deposited in distinct regions within each well. Plates were left to dry in the biosafety cabinet overnight, protected from light, and then washed once with PBS prior to cell seeding. Anti-mCherry synNotch ETV2-BFP or dual-lineage fibroblasts were seeded at $5–20 \times 10^4$ cells/cm² and cultured for three days prior to flow cytometry analysis or fixation and staining for VEGFR2.

### Staining

Flow Cytometry: cells were detached using TrypLE (Thermo Fisher) and washed once prior to incubation with fluorescently tagged antibodies in PBS + 5%FBS for 30 min–1 h at 4 °C. Following, cells were washed twice with PBS + 5%FBS and filtered through 35 μm cell strainer prior to analysis with ARIA II.

Following culture, cells were washed once with PBS, fixed with 4% paraformaldehyde or 10% ice-cold methanol for 10 min, and then washed 3× with PBS for 5 min each. Samples were stained immediately or further permeabilized with 0.1% Triton X-100 in PBS for 5–10 min and then washed 3× with PBS for 5 min. Cells were blocked for 1 h with 2% BSA at room temperature, then incubated with primary antibodies for 2 h at room temperature or overnight at 4 °C. Following three washes with PBS, samples were incubated with secondary antibodies for one hour at room temperature, then washed again prior to imaging directly or staining nuclei with NucBlue (15 min, Thermo Fisher) or Nuclear Mask Deep Red (30 min, Thermo Fisher). Samples on a coverslip were mounted with gold-antifade mounting solution (Thermo Fisher).

## Imaging/microscopy

Unless otherwise stated, a digital Microscope (Keyence BZ-X) was used to image experiments. Tiling was done with the built-in Keyence software. All images within individual experiments were taken with the same settings (Light strength, exposure, No LUT). BFP, GFP, mCherry, and miRFP signals were captured using the respective Filter cubes: BFP, GFP, TexasRed, Cy5-NX.

## RNA sequencing

For bulk RNA sequencing analysis, GFP or mCherry solid circle patterns were created via microcontact printing with 8 mm diameter stamps. $5 \times 10^4$ of the following cells were droplet seeded with and without the presence of their respective ligands: C3H parental (no-ligand only), anti-GFP/tTA synNotch that activates mCherry, anti-GFP/tTA synNotch that activates myoD and mCherry, anti-mCherry/Gal4 synNotch that activates ETV2 and BFP, C2C12 cell line (no-ligand only), and BEnd.3 cell line (no-ligand only). To ensure all cells were cultured on the activating ligand, cells were seeded as 50 µL droplets within the borders of the patterned ligands and allowed to adhere for 30 min before pipetting in the rest of the culture media. The same cell seeding strategy was performed for conditions without ligands. Cells were cultured for before total RNA extraction using miRNeasy kit per the manufacturer's protocol (Qiagen). An RNA cleanup kit was used to further purify/clean the samples (Zymo RNA Clean and Concentrator). RNA samples were then sequenced with an Illumina NovaSeq 6000 (Novogene Corporation Inc).

Fastqc files were trimmed with trimmomatic v0.39 using default settings. Trimmed fastQ files were aligned to GRCm38 reference genome supplemented with custom transgenic sequences using STAR v2.7.10b[87] with default parameters. Transcriptome alignments were quantified using featureCount[88] using the custom gene annotation file combining the GENCODE annotation file and transgenes. Gene counts were imported into R and differentially expressed genes were identified with DESEq2 v1.38.3[89] with $p_{adj} = 0.05$ as the threshold. For heatmap, Z-Score is calculated by (Gene expression value in sample of interest) − (Mean expression across all samples)/Standard Deviation. GO analysis was performed on the differentially expressed genes using the clusterProfiler v4.6.2[90] package.

## 10x single-nuclear RNA sequencing

For single-nuclear RNA sequencing analysis, patterns were prepared using the capillary fluidic device to contain either GFP in both inlets, mCherry in both inlets, or GFP/mCherry in one inlet each to generate a dual-ligand pattern. $3.3 \times 10^4$ dual-lineage fibroblasts were seeded in a 30 µL droplet on top of the patterns, allowed to attach for 30 min, prior to adding additional media. Cells were cultured for an additional 3 days before collection. To collect, cells were trypsinized and cells were lysed in IGEPAL CA630-containing lysis buffer for 7 min to isolate individual nuclei. Library construction was performed according to the manufacturer's protocol (10x Genomics single-cell 3′ v3.1 protocol). Briefly, after resuspension and counting, 16,000 GCs per experiment were

resuspended in the master mix and loaded (together with partitioning oil and gel beads), onto each lane of an 8-lane chip G to generate the gel bead-in-emulsion (GEMs). Reverse transcription was primed with an oligonucleotide carrying an Illumina TruSeq R1 read-sequencing primer, a 16 nucleotide 10x cell barcode, a 12 nucleotide UMI, and a 30 nucleotide anchored poly dT sequence. Full-length cDNA was amplified from heteroduplex RNA:cDNA using 12 cycles of PCR. The full-length cDNA was cleaned up on SPRIselect beads and QCed on Qubit and BioAnalyzer. One-fourth of the resulting ds cDNA was fragmented and prepared for sample index PCR, with 11 cycles of amplification. After QC, the libraries were pooled and submitted for sequencing on 2 lanes of a 10B 100 flowcell on the Illumina NovaSeqX sequencer, targeting a minimum read depth per cell of 25,000. Sequencing was performed at the UCSF CAT, supported by UCSF PBBR, RRP IMIA, and NIH 1S10OD028511-01 grants.

FASTQ files were processed with 10x Genomics' Cell Ranger analysis pipelines. The read count matrix generated by Cell Ranger was then analyzed using Seurat v5.0.2. 32,288 genes were detected across no ligand (7877 and 7745 cells tested for replicates), GFP pattern (10,539 and 11,923 cells tested for replicates), mCherry pattern (10,671 and 10,068 cells tested for replicates), and Dual Pattern (8710 and 16,266 cells tested for replicates). Cells that had unique feature counts with at least 700 genes but no more than 7000 genes and cells that had <55% mitochondrial counts were filtered and normalized based on the feature expression and total expression of each cell. The normalized expression data were then used for subsequent analysis.

Principal component analysis was performed after merging replicates and integrating all the conditions. Highly variable genes in each sample after linear transformation and the first 30 PC scores were used for tSNE analysis to cluster the cells into 12 groups (FindNeighbors and FindClusters functions implemented in the Seurat package, dims = 30, resolution = 0.4). The marker genes of each cluster were identified using FindAllMarkers or FindMarkers function with default parameters. Clusters were annotated using signature genes and DAVID pathway analysis to identify fibroblast-, muscle-, or endothelial-like cell types across the different conditions. tSNE clusters that were enriched in proliferation, extracellular matrix, or EGF pathways were identified as fibroblasts. Clusters that were enriched in lineage-specific markers, muscle or angiogenesis pathways, were used to identify muscle- and endothelial-like clusters, respectively.

A pseudobulk method was applied to investigate gene expression among different conditions at the population level. Specifically, the raw gene counts of each sample were extracted after filtering. The counts were then aggregated to the sample level and the expression of genes of interest including transgenes were examined across conditions.

## Statistics

Individual data points in graphs represent distinct samples. Statistics were calculated in Prism, using Unpaired T-test two-tailed or one-way Anova between groups and Tukey's multiple comparisons tests. $*p < 0.05$, $**p = <0.01$, $***p = <0.001$, $****p = <0.0001$ with a 95% confidence interval.

## Reporting summary

Further information on research design is available in the Nature Portfolio Reporting Summary linked to this article.

## Data availability

The source data file at the following link contains the quantified data points for each data graph in the figures. https://doi.org/10.6084/m9.figshare.25648026. The bulk and single-nuclei RNA sequencing data generated in this study have been deposited in the NIH GEO database under accession code GSE269404. Plasmids generated in this study are available on Addgene. Source data are provided with this paper.

## Code availability

The ImageJ macro used for the generation of scrambled pixel images for Fig. 3 is provided in the supplemental information file.

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

## Acknowledgements

The authors acknowledge Marion Johnson and all the members of the Morsut, McCain, Li, Khademhosseini Lab for insightful discussions and suggestions on the project. The authors acknowledge their family and friends that support them always and in particular for the times of this work that took place during the COVID-19 pandemic. Research reported in this publication was supported by NIGMS of the National Institutes of Health under award number R35GM138256 (L.M.); the National Science Foundation award number CBET-2145528 Faculty Early Career Development Program (L.M.); NSF RECODE from CBET-2034495 (M.L.M., L.M.); USC Department of Stem Cell Biology and Regenerative Medicine Startup Fund (L.M.); Wellcome Trust, Leap HOPE (L.M.), Viterbi Center for CIEBOrg (L.M., M.L.M.). T.H. acknowledges support from the Ruth L. Kirschstein National Research Service Award T32HL069766 and the UCLA Eli and Edythe Broad Center of Regenerative Medicine and Stem Cell Research, Research Award. Fellowships for students: CIRM fellowship for S.D., Fellowships from BME Department for the first year for M.G., N.C., and S.D. Grace True, Finacy Jin for technical support for the project. S.L. acknowledges the support of the Innovation Award from the UCLA Eli and Edythe Broad Center of Regenerative Medicine and Stem Cell Research, and grants (GM143485 and NS126918) from the National Institute of Health. GFP structures used in Figs. 1A, F and 2A, D created with BioRender.com. Marvin was used for drawing and displaying chemical structures, substructures, and reactions, Marvin 19.7.0, 2019, ChemAxon (http://www.chemaxon.com). This project has been made possible in part by grant number 2023-332386 from the Chan Zuckerberg Initiative DAF, an advised fund of Silicon Valley Community Foundation.

## Author contributions

M.G., T.H., T.M., S.D., A.R.M., N.C., S.L., M.L.M., and L.M. designed the experiments. M.G., T.H., T.M., S.D., N.P., R.E.L., J.S., and B.J. performed the experiments. M.G., T.H., T.M., S.D., R.E.L., and B.J. analyzed the data. M.G., T.H., T.M., S.D., A.K., S.L., M.L.M., and L.M. contributed to data interpretation and discussion. M.G., T.H., T.M., S.D., S.L., M.L.M., and L.M. wrote the manuscript. L.M., M.L.M., S.L., and A.K. acquired funding for the project.

## Competing interests

The technology transfer office of USC with the authors have filed patent disclosures with the technology described here; LM is an inventor on a previous synNotch patent for applications in cancer cell therapy licensed to Gilead; M.L.M. is an inventor on a patent on gelatin hydrogels licensed to Emulate. The remaining authors declare no competing interests.
