## [Peer Review File · Nature Communications]

Reviewers' Comments:

Reviewer #1:

Remarks to the Author:

The manuscript by McCain / Morsut and their team reports an elegant study based on synthetic Notch receptors to spatially control synthetic gene activation and thus lineage patterning in different multicellular contexts. Synthetic Notch receptors are an exciting synthetic biology tool that can be genetically engineered into mammalian cells to detect signals presented by surrounding environments such as neighboring cells and respond by activating prescribed transcriptional programs. Previous studies have mainly focused on using cell-presented ligands to activate synthetic Notch receptors. In this manuscript, the authors have significantly extended the novel applications of this powerful tool to tissue engineering applications. Overall, I find the data presented in the manuscript of high quality and well organized and presented. A few comments below that I hope the authors can address in their revised manuscript; I hope they will be useful for the authors to further extend the impact of this manuscript.

Some of the data or investigations can be strengthened; this is particularly true for the last part of this work, which I find most exciting and might benefit from additional data. The authors here aimed to demonstrate spatially controlled co-transdifferentiation of fibroblasts to endothelial cell precursors and skeletal muscle precursors, by using dual-lineage fibroblasts expressing two independent synNotch receptors and culturing these cells on a surface with the two synthetic cognate ligands patterned via a microfluidic device. The authors only used a very limited number of lineage markers to show the co-transdifferentiation. I would suggest the authors to consider first using bulk RNA-seq to characterize the effects of overexpression of MyoD and ETV2 on transdifferentiation of fibroblasts to endothelial cell precursors and skeletal muscle precursors, respectively. Then the authors could consider applying scRNA-seq to further investigate whether their microfluidic ligand patterning tool indeed could lead to lineage bifurcation towards endothelial cell precursors and skeletal muscle precursors in the same culture.

In Fig. 4D, for the image showing cells on GFP & mCherry intersection regions, I am wondering why most cells still appear to only express either BFP or miRFP. Maybe I am wrong here. Nonetheless, maybe the authors should quantify the percentage of cells co-expression BFP / miRFP. Related to this, could the authors comment future possibilities to add cross-inhibitory modules in the synthetic genetic network to promote lineage bifurcation even when two synthetic cognate ligands are patterned in the same area.

Maybe the authors could add some discussions about how heterogenous synthetic gene activation is at the single cell level based on synthetic Notch receptors. How such gene expression heterogeneity is compared with other conventional tools, such as based on exogenous morphogen signals.

Reviewer #2:

Remarks to the Author:

The manuscript by Garibyan et al aims to develop novel approaches to control spatial patterning of cells in vitro. By using the synNotch system, the authors tested various ways to tether synNotch ligands to culture substrates with defined spatial patterns, which can induce gene expression in synNotch receivers. Furthermore, the authors demonstrated that different synNotch ligand-receptor pairs enable orthogonal gene expression controls, which can be used to induce the transdifferentiation of fibroblast cells into endothelial and skeletal muscle cells in a spatially defined manner. Although the synNotch synthetic design and its application were previously published, this manuscript is the first demonstration that physically tethering synNotch ligands outside cells can control gene expression in synNotch receivers, and this system can be integrated with other established tissue engineering tools, such as conjugation onto synthetic extracellular matrix and 3D printing. This technology will provide the tissue engineering field with additional

modes to control tissue patterning, and therefore will be relevant to the broader tissue engineering community.

Below are minor concerns and revisions that I recommend to strengthen the conclusions of the paper:

1. Figure 2H/S2J - Mention of the live/dead staining in the hydrogel setup within the main text would be appreciated. The authors may also want to quantify cell viability with the data shown in Figure S2J, to demonstrate the compatibility of the setup for supporting cell survival.
2. Figure 4 - Authors should perform some single cell quantification of the BFP and mCherry signal for each cell in the different substrate regions. This quantification would strengthen the author's argument on orthogonal control of gene expression in the same cell.
3. Figure 5A - Since one of the readouts of this assay is alignment of cells along the axis, it may be useful to demonstrate that the geometry observed is due to MyoD expression, rather than physical constraints imposed by the grooves within the gelatin. Perhaps one could measure this in the α -Actinin negative cells by using an actin stain, which might show parallel stress fibers if cells are aligned.
4. Methods (Data quantification) - A brief description of how the used toolboxes work (ex. OrientationJ) would be appreciated, for those not familiar with these plugins.

Reviewer #3:

Remarks to the Author:

In their submitted manuscript, Garibyan et al. utilize surface-patterned biomaterials in conjunction with engineered fibroblasts to spatially control gene expression through SynNotch. This is one of the most visually striking manuscripts that I have seen presented in the biomaterials literature. The paper is well written and is nicely complemented by a collection of gorgeous imagery and thoughtful figure design.

Employing a small army of biomaterial/cell engineering techniques, the authors convincingly demonstrate that engineered fibroblasts interfacing with patterned SynNotch ligand can be used to drive changes in 2D gene expression. Most examples are confined to fluorescent proteins: immobilizing GFP to turn on expression of intracellular mCherry, or immobilizing mCherry to drive expression of intracellular BFP. Notably, however, are the studies in Figures 6 and 7 which induce myogenic endothelial differentiation from fibroblasts via patterned SynNotch and expression of master regulators ETV2 and MyoD.

My primary concerns were related to the manuscript's misrepresentation of the prior literature, both in the main text and abstract, which in turn limits the submitted paper's novelty. Specifically, Ref. 27 was published earlier this year by the Brunger lab that, among other things, created biomaterial surfaces patterned with different concentrations of immobilized GFP to drive where and how much mCherry is expressed via SynNotch. This is in direct contrast of the authors statement in main text paragraph 3 that "none of these approaches attempted patterning of ligands, a feature that would enable spatial control of synNotch activation and therefore gene expression." I do think that the presented work is sufficiently novel to warrant potential publication in Nature Communications, but that a full and proper discussion must be included highlighting the Brunger efforts and how those presented differ. I note that Brunger's paper was available online on March 29, 2023, and that Garibyan's preprint was posted to bioRxiv on May 20, 2023 – presumably in response to the Brunger lab publication.

Noticeably absent is also discussion of controlled multimaterial bioprinting, where different cell types can be spatially deposited in 2D/3D arrangements. The resolutions of these alternative techniques appear to be as good if not better than what is achieved in Figure 7. In this light, and in consideration of optogenetic methods which have been demonstrated with subcellular

resolutions, are comments like “these generalizable technologies provide users with unprecedented abilities to dictate spatial patterning of gene expression in multicellular constructs” valid?

The manuscript would also benefit from a realistic description of the strategy’s limitations. Though I am not in this field, some that immediately come to mind are: limitations to 2D (at least in present study), potential patterning resolution, a requirement for both engineered materials and engineered cells, the limited collection of master regulators that could drive major phenotypic changes upon SynNotch activation, limited cell types that will accept the SynNotch machinery, leaky activation, and likely others. An honest discussion of these would benefit the manuscript tremendously.

New plasmids should be deposited on Addgene prior to acceptance and made available to the community upon publication.

Images in Figure 4D should be substituted with those at higher resolutions, matching that presented throughout the rest of the paper.

Response to Reviewers

We sincerely thank the three reviewers for their time in reviewing our manuscript and providing invaluable insights that have helped us produce a better version of the paper.

Briefly, in this manuscript, we demonstrated novel ways to spatially control gene expression in mammalian cells with a combination of materials engineering and cell engineering via synNotch. We first showcase this with a number of materials and reporter genes in 2D and 3D. We then demonstrate the capacity to spatially control cell differentiation programs by micropatterning the ligands and subsequent co-trans-differentiating mouse embryonic fibroblasts towards endothelial precursors and skeletal muscle precursors.

Reviewer comments focused on 3 themes (more details in the point-by-point response):

(i) Suggestions to improve the characterization of cell differentiation with sequencing technologies: We performed bulkRNA sequencing for cells with single synNotch receptors on materials with single ligands, and performed single-cell sequencing for cells with two synNotch receptors on materials patterned with two ligands. These data are now in Fig. 5-7, which characterize differentiation far beyond the few markers that were used in the first submission, now giving a much more comprehensive picture.

(ii) Helpful technical comments: We addressed these with some new experiments, analysis, and/or explanations.

(iii) Suggestions for improving the Introduction and Discussion: We added more references and comparisons to complementary technologies, which gives more context to our contribution.

With the modifications summarized here and detailed below, we believe that the manuscript is much stronger, and that we have adequately addressed the reviewers' concerns.

Notes:

- *In the responses below, reviewer comments are in black, **our responses are in red** and a **summary of changes to the manuscript are in blue**; we also inserted the new additional figures that have been added in line with the response letter here;*
- *In the revised manuscript files, new figures panels are indicated with a red boundary; in the text, **new text is in red**.*

Reviewer #1 comments, and our responses

Note: reviewer comments are in black, our responses are in red and a summary of changes to the manuscript are in blue

The manuscript by McCain / Morsut and their team reports an elegant study based on synthetic Notch receptors to spatially control synthetic gene activation and thus lineage patterning in different multicellular contexts. Synthetic Notch receptors are an exciting synthetic biology tool that can be genetically engineered into mammalian cells to detect signals presented by surrounding environments such as neighboring cells and respond by activating prescribed transcriptional programs. Previous studies have mainly focused on using cell-presented ligands to activate synthetic Notch receptors. In this manuscript, the authors have significantly extended the novel applications of this powerful tool to tissue engineering applications. Overall, I find the data presented in the manuscript of high quality and well organized and presented.

We thank the reviewer for their enthusiasm for our work!

A few comments below that I hope the authors can address in their revised manuscript; I hope they will be useful for the authors to further extend the impact of this manuscript.

1. Some of the data or investigations can be strengthened; this is particularly true for the last part of this work, which I find most exciting and might benefit from additional data. The authors here aimed to demonstrate spatially controlled co-transdifferentiation of fibroblasts to endothelial cell precursors and skeletal muscle precursors, by using dual-lineage fibroblasts expressing two independent synNotch receptors and culturing these cells on a surface with the two synthetic cognate ligands patterned via a microfluidic device. The authors only used a very limited number of lineage markers to show the co-transdifferentiation.

I would suggest the authors to consider first using bulk RNA-seq to characterize the effects of overexpression of MyoD and ETV2 on transdifferentiation of fibroblasts to endothelial cell precursors and skeletal muscle precursors, respectively.

We thank the reviewer for the important suggestion. In response, we completed bulk RNA sequencing of the following conditions to more thoroughly evaluate the transcriptome upon synNotch-triggered differentiation towards myogenic and endothelial lineages:

- C3H fibroblasts on fibronectin
- anti-GFP/mCherry synNotch C3H fibroblasts with and without GFP ligand
- anti-GFP/myoD synNotch C3H fibroblasts with and without GFP ligand
- anti-mCherry/ETV2 synNotch C3H fibroblasts with and without mCherry ligand
- C2C12 myotubes (positive control for myogenic differentiation)
- Bend.3 endothelial cells (positive control for endothelial differentiation)

We then performed principal component analysis (PCA), hierarchical clustering, and GO-term functional analysis. Accordingly, for both myogenic and endothelial differentiation, the results demonstrate that the presence of the correct synNotch receptor-ligand pair leads to lineage-specific changes in the cells, pushing them towards myogenic or endothelial lineages and closer to the positive control cells (C2C12 or Bend.3, respectively).

We also want to clarify that we are not claiming to achieve complete differentiation to mature cell types; this would likely require modification of the culture medium (e.g., supplementation with growth factors) and/or extension of the culture time. Instead, we show

induction towards myogenic or endothelial lineages after just 3 days of culture in basal medium. We clarified this concept and associated language throughout the manuscript.

Revisions to the manuscript:

- Bulk RNA sequencing of myogenic transdifferentiation, including hierarchical clustering, volcano plot, and GO-term analysis (Fig. 5C-E; pasted below)
- Bulk RNA sequencing of myogenic transdifferentiation, including hierarchical clustering and volcano plot (Fig. 6D-E; pasted below)
- PCA plot of all cell types showing progression towards myogenic and endothelial lineages (Fig. S7G; pasted below)
- Updated description in the text of these data

Fig. 5C-E:

Fig. 6D-E:

Fig. S7G:

2. Then the authors could consider applying scRNA-seq to further investigate whether their microfluidic ligand patterning tool indeed could lead to lineage bifurcation towards endothelial cell precursors and skeletal muscle precursors in the same culture.

We thank the reviewer for the inspiring comment to further investigate dual differentiation on dual patterned ligands. In response, we performed single-nuclei RNA sequencing analysis of the dual lineage cell line after culture for 3 days in basal medium on four microfluidic patterns: no ligands, GFP-only lanes, mCherry-only lanes, and interdigitating mCherry/GFP lanes. The analysis showed that, on the dual mCherry/GFP pattern, the cells do indeed undergo lineage bifurcation towards cells with endothelial cell precursor signatures and cells with skeletal muscle precursor signatures, in the same culture.

This analysis also allowed us to investigate the other conditions: (i) on no ligands, clusters were identified as mainly cells with fibroblast signatures; (ii) on mCherry-only lanes, we observed the formation of a new cluster, representing an endothelial-like cell population, that is overexpressing marker genes like KDR, CDH5; (iii) on GFP-only lanes, we observe two new clusters, representing myogenic-like cell populations that are overexpressing myogenic-related genes. Interestingly, on the dual-ligand patterns, both the GFP-induced and the mCherry-induced clusters are present, and we also observed the emergence of a new sub-muscular cluster, unique to the co-differentiation platform. We described these additional experiments in the Methods section “10x Single-Nuclear RNA Sequencing”, Figure 7D, in the main text, and the methods in the section “10x Single-Nuclear RNA Sequencing”. And we have submitted the dataset in the repository NIH GEO, which we think are going to be helpful for further analysis from the interested parties.

Revisions to the manuscript:

- Description of new experiments and data in Methods, Main text, and Discussion
- Single cell sequencing results in Fig. 7D-F , Fig. S10A,B and Table S1,2
- New paragraph in Discussion to discuss relevance of the new data.

Fig. 7D-F:

Fig. S10A,B:

Table S1 and S2:

Rank	Cluster 0	Cluster 1	Cluster 2	Cluster 3
1	Cell Cycle	Differentiation	Extracellular	Synapse
2	Chromosome	Calmodulin Binding	Heparin Binding	Cell Membrane
3	Mitosis	Cardiomyopathy	Growth Factor	PH
4	Microtubule Cytoskeleton	Muscle Contraction	Migration	Rho GTPase
5	Nucleus	Extracellular	Focal Adhesion	Ion Channel
6	Mitotic Assembly	Sarcoplasmic Reticulum	Innate Immunity	MAM
7	Microtubule	Actin Binding	Calcium binding	Presynaptic membrane
8	Nucleotide Binding	Troponin	Pathways	Cardiomyopathy
9	Mitotic Cell Cycle	Transmembrane	Collagen	SH3
10	Isopeptide Bond	HLH	kinase signaling	Angiogenesis/ VEGF
	(14) EGF		(11) EGF (13) MMP	
Assigned Identity:	Fibroblast-like Dividing	Muscle-like	Fibroblast-like	Endothelial-like
Cluster 4	Cluster 5	Cluster 6	Cluster 7	Cluster 8
Postsynaptic	Ribosome / Translation	Actin Binding	Ribosome	Cell Cycle
Differentiation	Muscle Protein	Cardiomyopathy	Muscle Protein	Chromosome
Membrane	Troponin	Differentiation	Cytoplasm	Mitosis
PH domain	Cytoplasm	Sarcoplasmic Reticulum	Muscle Contraction	Cell Cycle
Signaling Pathways	Cardiac muscle contraction	Fibronectin	Muscle Contraction	Cytoskeleton
Fibronectin	Muscle Contraction	PDZ	Troponin	Cell Cycle
SUSHI	EF-hand	PH domain	Sarcoplasmic Reticulum	ATP binding
Extracellular Matrix	Ribosomal	Calmodulin	Cardiomyopathy	Mitosis
EGF	Carboxylesterase	Muscle Proteins	Striated Muscle	Cyclin
Extracellular	Chemotaxis	Troponin	EF-hand	Isopeptide
Fibroblast	Muscle-like	Muscle-like	Muscle-like	Fibroblast-like Dividing
Cluster 9	Cluster 10	Cluster 11		
Extracellular Secreted	ribosome	Extracellular secreted		
Cell Cycle	cytoplasm	Response to virus		
Extracellular matrix	ribosome	Extracellular matrix		
Chromosome	Isopeptide	Immune Signaling		
Cytoskeleton	Spliceosome	Furin		
Heparin-binding	Signal Recognition Particle	TSP1		
EGF-like	EF-hand1	GFP binding		
Signal Pathway	Cell Cycle	Response to IFN β		
PDGF	Oxidative Stress	Oxidative Stress		
EGF-like	Antioxidant	TIR domain		
Fibroblast-like	Fibroblast-like	Fibroblast-like		

RANK	Muscle 1 vs. Muscle 2	Muscle 1 vs. Muscle 3
	UP	DOWN
1	Ribosome	Cardiomyopathy
2	Ribosomal	PH Domain
3	Cytoplasm	Actin Binding
4	Extracellular Secreted	Sarcoplasmic Reticulum
5	Isopeptide Bond	PDZ
6	Collagen/FDGR binding	RhoGTP
7	Signaling	Differentiation
8	Iron	Muscle Protein
9	Antioxidant	Fibronectin
10	Signaling Pathway	SH3 Domain
	(14) EGF	
	UP	DOWN
	Cell Projection	ribosome
	Cell Differentiation	cell cycle
	PH domain	ribosomal
	Calmodulin binding	Chromosome
	Hormone	Cytoskeleton
	Rap1 Signaling	Isopeptide Bond
	Transcription	Mitosis
	Cell Junction	Growth factor
	PDZ	Microtubule
	SH3	EF-hand
Muscle 1 vs. Muscle 4	Muscle 2 vs. Muscle 3	
UP	DOWN	
Developmental Protein	ribosome	
Actin Binding	cell cycle	
Calmodulin Binding	ribosome	
Extracellular Matrix	extracellular secreted	
Cardiomyopathy	chromosome	
Glycoprotein	actin binding	
PDZ	extracellular matrix	
Sarcoplasmic Reticulum	Isopeptide bond	
Fibronectin/IgG	EF-hand	
PH domain	Microtubule	
	UP	
	PH domain	
	Cardiomyopathy	
	Actin Binding	
	PDZ	
	SH3	
	extracellular secreted	
	Isopeptide	
	extracellular matrix	
	mitosis	
	mitosis	
	G8 Domain	
	SAM Domain	
	immune signaling	
	Hormone	
	microtubule	
Muscle 4 vs. Muscle 2	Muscle 4 vs. Muscle 3	
UP	DOWN	
Ribosome	actin binding	
Cell Cycle	differentiation	
Extracellular Secreted	Fibronectin	
Differentiation	PH domain	
PH domain	PDZ	
Fibronectin	cardiomyopathy	
Actin Binding	muscle protein	
Chromosome	cell junction	
Rho GTPase	SH3 domain	
(13) Muscle Contraction		
	UP	
	Extracellular matrix	
	actin binding	
	sarcoplasmic reticulum	
	extracellular secreted	
	muscle contraction	
	muscle protein	
	signal	
	protein response	
	cardiac contraction	
	peptide signaling	

3. In Fig. 4D, for the image showing cells on GFP & mCherry intersection regions, I am wondering why most cells still appear to only express either BFP or miRFP. Maybe I am wrong here. Nonetheless, maybe the authors should quantify the percentage of cells co-expression BFP / miRFP.

We thank the reviewer for the helpful comments, as highlighting the dual-activation of cells at the GFP-mCherry intersection is an important message of the figure. We agree that it is challenging to see the overlap of BFP and miRFP by eye in Fig. 4C, owing to the blue and violet colors (respectively) used to represent the fluorescent proteins. In Fig. S5G, we show the same image overlaid with a mask that indicates regions that are BFP+/miRFP+ with yellow outlines. These images show more clearly that many cells express both reporters at the GFP-mCherry intersection, indicative of dual activation.

As suggested by the reviewer, we also quantified the number of cells expressing fluorescent reporters (BFP, miRFP) on the four different regions of the pattern (no ligand, GFP only, mCherry only, GFP-mCherry intersection) with image analysis (new Fig. 4D). On single ligands (GFP or mCherry), ~60-70% of cells express the cognate reporter (miRFP or BFP, respectively). On the intersections (GFP and mCherry), ~50% of cells express both reporters (BFP and miRFP). The levels of activation shown in this figure (Fig. 4D) are similar to the percentage of cells that expressed both reporters when cultured on substrates coated uniformly with both ligands and analyzed via flow cytometry (Fig. S5B).

Revisions to the manuscript:

- Quantification of BFP and miRFP expression on GFP-mCherry perpendicular line pattern (new Fig. 4D)

- Reorganization of Fig. S5B
- Updated description in the text of these data

Fig. 4D and Fig. S5B:

4. Related to this, could the authors comment future possibilities to add cross-inhibitory modules in the synthetic genetic network to promote lineage bifurcation even when two synthetic cognate ligands are patterned in the same area.

We thank the reviewer for this comment that stimulated a good discussion in our group; we have added a few comments on future possibilities related to this in the Discussion section. We also shared new data that we collected regarding dual differentiation on a substrate that presents both ligands, where neither differentiation seems to be able to proceed, suggesting that myoD and ETV2 are attempting to promote their own program, but failing in the presence of the other, which is different than our results with fluorescent reporters. This result is shown in the new Fig. S9A.

Revisions to the manuscript:

- New paragraph in Discussion
- New data in Fig. S9A and associated discussion in the Main text

Fig. S9A:

5. Maybe the authors could add some discussions about how heterogeneous synthetic gene activation is at the single cell level based on synthetic Notch receptors. How such gene expression heterogeneity is compared with other conventional tools, such as based on exogenous morphogen signals.

We thank the reviewer for the interesting comment. We have added a new paragraph in the Discussion where we collated the data we have collected on single cell activation of synNotch and differentiation, and comment on the heterogeneity that we observe. It is harder to compare our results directly to other methods of differentiation, especially since we do not claim to drive cells towards highly mature phenotypes. Reaching higher levels of maturity will likely require supplemented differentiation media and/or longer time in culture, which is outside the scope of the current paper.

Revisions to the manuscript:

- New paragraph in the discussion:

In terms of heterogeneity, we observed bimodal and therefore incomplete activation of synNotch by ligands presented by materials, similar to other studies that have presented synNotch ligands from cells or other materials. Across all materials we tested, we found that synNotch activation reached a plateau in response to increasing ligand concentration, beyond which synNotch activation did not increase. Thus, we likely reached the saturation point of ligand presentation by the material and synNotch signaling itself seems to be the main factor limiting activation. We correspondingly observed an imperfect differentiation efficiency, which is likely a compounded effect of the heterogeneity of synNotch activation and the known heterogeneity of transcription-factor-mediated differentiation, especially at the early time points that we investigated in this study. These are major limitations of synNotch but will continue to improve as the technology evolves.

Reviewer #2 comments, and our responses

Note: reviewer comments are in black, our responses are in red and a summary of changes to the manuscript are in blue

The manuscript by Garibyan et al aims to develop novel approaches to control spatial patterning of cells in vitro. By using the synNotch system, the authors tested various ways to tether synNotch ligands to culture substrates with defined spatial patterns, which can induce gene expression in synNotch receivers. Furthermore, the authors demonstrated that different synNotch ligand-receptor pairs enable orthogonal gene expression controls, which can be used to induce the transdifferentiation of fibroblast cells into endothelial and skeletal muscle cells in a spatially defined manner. Although the synNotch synthetic design and its application were previously published, this manuscript is the first demonstration that physically tethering synNotch ligands outside cells can control gene expression in synNotch receivers, and this system can be integrated with other established tissue engineering tools, such as conjugation onto synthetic extracellular matrix and 3D printing. This technology will provide the tissue engineering field with additional modes to control tissue patterning, and therefore will be relevant to the broader tissue engineering community.

Below are minor concerns and revisions that I recommend to strengthen the conclusions of the paper:

1. Figure 2H/S2J - Mention of the live/dead staining in the hydrogel setup within the main text would be appreciated. The authors may also want to quantify cell viability with the data shown in Figure S2J, to demonstrate the compatibility of the setup for supporting cell survival.

Thank you for the comment. We added a statement mentioning the live/dead staining as well as quantification to demonstrate our hydrogel platform supports cell viability.

Revisions to the manuscript:

- *New text in Main section: "GelMA hydrogels maintain encapsulated cell viability for at least seven days maintaining ~90% viability when quantified via Live/Dead staining (Fig. S2J)"*

2. Figure 4 - Authors should perform some single cell quantification of the BFP and mCherry signal for each cell in the different substrate regions. This quantification would strengthen the author's argument on orthogonal control of gene expression in the same cell.

We thank the reviewer for the helpful comments (Reviewer #1 had a similar comment), as highlighting the dual-activation of cells at the GFP-mCherry intersection is an important message of the figure. We agree that it is challenging to see the overlap of BFP and miRFP by eye in Fig. 4C, owing to the blue and violet colors (respectively) used to represent the fluorescent proteins. In Fig. S5G, we show the same image overlaid with a mask that indicates regions that are BFP+/miRFP+ with yellow outlines. These images show more clearly that many cells express both reporters at the GFP-mCherry intersection, indicative of dual activation.

As suggested by the reviewer, we also quantified the number of cells expressing fluorescent reporters (BFP, miRFP) on the four different regions of the pattern (no ligand, GFP only, mCherry only, GFP-mCherry intersection) with image analysis (new Fig. 4D). On single ligands (GFP or mCherry), ~60-70% of cells express the cognate reporter (miRFP or BFP, respectively). On the intersections (GFP and mCherry), ~50% of cells express both reporters (BFP and miRFP). The levels of activation shown in this figure (Fig. 4D) are similar to the

percentage of cells that expressed both reporters when cultured on substrates coated uniformly with both ligands and analyzed via flow cytometry (Fig. S5B).

Revisions to the manuscript:

- Quantification of BFP and miRFP expression on GFP-mCherry perpendicular line pattern (new Fig. 4D)
- Reorganization of Fig. S5B
- Updated description in the text of these data

Fig. 4D and Fig. S5B:

3. Figure 5A - Since one of the readouts of this assay is alignment of cells along the axis, it may be useful to demonstrate that the geometry observed is due to MyoD expression, rather than physical constraints imposed by the grooves within the gelatin. Perhaps one could measure this in the α -Actinin negative cells by using an actin stain, which might show parallel stress fibers if cells are aligned.

Thank you for the suggestion. We did not intend to imply that cell alignment is induced by MyoD expression; we expect that cell alignment is primarily driven by the physical constraints of the micromolded ridges and thus both activated and non-activated cells should align to the micromolded ridges. However, only cells on hydrogels with GFP should also activate MyoD.

As such, we agree with the reviewer that it is useful to quantify the alignment of all cells (non-activated and activated). Instead of staining actin filaments, as suggested by the reviewer, we stained and quantified the alignment of nuclei (another universal marker) in fibroblasts expressing anti-GFP synNotch that activates MyoD, cultured on isotropic or micromolded gelatin hydrogels, with or without GFP. As shown in the new Fig. S6B, nuclei on micromolded hydrogels were significantly more aligned than nuclei on isotropic hydrogels, regardless of MyoD expression, as expected. We also found a slight but non-significant increase in the alignment of nuclei for cells cultured on micromolded gelatin hydrogels with GFP compared to micromolded gelatin hydrogels without GFP, suggesting MyoD activation may cause a modest improvement in cell alignment, likely due to cell fusion.

Although not requested by the reviewer, we also completed additional experiments to compare the myogenic index of cells in all four conditions, shown in the updated Fig. 5B.

Revisions to the manuscript:

- Quantification of nuclei alignment (Fig. S6B) and myogenic index (Fig. 5G) for fibroblasts with anti-GFP MyoD synNotch on featureless/micromolded gelatin hydrogels with/without GFP
- Updated description in the text of these data

Fig. S6B and Fig. 5G:

4. Methods (Data quantification)

A brief description of how the used toolboxes work (ex. OrientationJ) would be appreciated, for those not familiar with these plugins.

We expanded the paragraph in the Methods section describing how ImageJ/OrientationJ and MATLAB were used to measure the Orientation Order Parameter of myotubes, including the addition of new references describing the underlying calculations.

Revisions to the manuscript:

- Updated Methods section

Reviewer #3 comments, and our responses

Note: reviewer comments are in black, our responses are in red and a summary of changes to the manuscript are in blue

In their submitted manuscript, Garibyan et al. utilize surface-patterned biomaterials in conjunction with engineered fibroblasts to spatially control gene expression through SynNotch. This is one of the most visually striking manuscripts that I have seen presented in the biomaterials literature. The paper is well written and is nicely complemented by a collection of gorgeous imagery and thoughtful figure design.

We thank the reviewer for their positive evaluation of our work and figures!

Employing a small army of biomaterial/cell engineering techniques, the authors convincingly demonstrate that engineered fibroblasts interfacing with patterned SynNotch ligand can be used to drive changes in 2D gene expression. Most examples are confined to fluorescent proteins: immobilizing GFP to turn on expression of intracellular mCherry, or immobilizing mCherry to drive expression of intracellular BFP. Notably, however, are the studies in Figures 6 and 7 which induce myogenic endothelial differentiation from fibroblasts via patterned SynNotch and expression of master regulators ETV2 and MyoD.

1. My primary concerns were related to the manuscript's misrepresentation of the prior literature, both in the main text and abstract, which in turn limits the submitted paper's novelty. Specifically, Ref. 27 was published earlier this year by the Brunger lab that, among other things, created biomaterial surfaces patterned with different concentrations of immobilized GFP to drive where and how much mCherry is expressed via SynNotch. This is in direct contrast of the authors statement in main text paragraph 3 that "none of these approaches attempted patterning of ligands, a feature that would enable spatial control of synNotch activation and therefore gene expression." I do think that the presented work is sufficiently novel to warrant potential publication in Nature Communications, but that a full and proper discussion must be included highlighting the Brunger efforts and how those presented differ. I note that Brunger's paper was available online on March 29, 2023, and that Garibyan's preprint was posted to bioRxiv on May 20, 2023 – presumably in response to the Brunger lab publication.

We thank the reviewer for the opportunity to further clarify the features of our system in comparison to other similar technologies in the field. We have highlighted in the revised manuscript the technology from the Brunger/Lim Lab in the introduction, and compared it to ours in the discussion.

Revisions to the manuscript:

- Modified the abstract
- Removed the sentence: "none of these approaches attempted patterning of ligands, a feature that would enable spatial control of synNotch activation and therefore gene expression."
- New section in Introduction that reads as follows:

More recently, an approach to specifically activate synNotch pathways from culture surfaces was developed under the acronym MATRIX (ref27). In this approach, surfaces are functionalized with antibodies (e.g. GFP-TRAP) that capture soluble synNotch ligands (e.g.

GFP), which can then activate synNotch receptors (e.g. anti-GFP synNotch) to regulate CRISPR-based transcriptome modifiers, modulate inflammatory niches, and mediate stem cell differentiation in receiver cells. Wedge-shaped culture inserts were also used to functionalize surfaces with coarse spatial control. However, whether synNotch ligands can be directly conjugated to a range of natural or synthetic biomaterials to activate synNotch, and whether this approach could be extended to pattern gene expression and/or differentiation and co-differentiation of multiple cell fates within the same culture with micron scale precision, has not been shown to date.

2. Noticeably absent is also discussion of controlled multimaterial bioprinting, where different cell types can be spatially deposited in 2D/3D arrangements. The resolutions of these alternative techniques appear to be as good if not better than what is achieved in Figure 7. In this light, and in consideration of optogenetic methods which have been demonstrated with subcellular resolutions, are comments like “these generalizable technologies provide users with unprecedented abilities to dictate spatial patterning of gene expression in multicellular constructs” valid?

We thank the reviewers for this comment and agree that our language was too strong, so we have reduced the claim in the specific comment indicated by the reviewer. We also agree that multi-material bioprinting is a complementary technology worthy of discussion. Thus, we expanded our discussion of optogenetic approaches and added a new discussion of multi-material bioprinting. Each of these technologies, including ours, has their own strengths and weaknesses, and we believe that we have now provided a more balanced view. We also added a point of enthusiasm regarding the future integration of these technologies.

Revisions to the manuscript:

- Edited the sentence highlighted by the reviewer
- Added a new paragraph in the discussion:

Patterning multi-lineage tissues with spatial control has also been achieved with multi-material extrusion bioprinting. In a recent example, human stem cells were engineered with doxycycline-inducible transcription factors for endothelial or neural cell fates. Wildtype or engineered cells were embedded in individual bioinks and merged into a tri-layer filament before extrusion through a nozzle in user-defined patterns. Doxycycline and differentiation media were then added to induce the co-differentiation of neural stem cells, endothelial cells, and neurons. Although this is a powerful approach for multi-lineage tissue engineering, it does have its own limitations, such as the reliance on diffusion-limited soluble factors for differentiation, restrictions on spatial resolution imposed by the nozzle, and a somewhat limited library of printable materials. However, we envision many synergistic opportunities for bioprinting and synNotch technologies to be used together by, for example, functionalizing printable bioinks with synNotch ligands. Optogenetic technologies, as mentioned above, can also be integrated to add more temporal control of cell phenotype. Overall, the ongoing integration of synthetic biology, biomaterials, and microfabrication technologies will further advance the capabilities for tissue engineering.

3. The manuscript would also benefit from a realistic description of the strategy's limitations. Though I am not in this field, some that immediately come to mind are: limitations to 2D (at least in present study), potential patterning resolution, a requirement for both engineered materials and engineered cells, the limited collection of master regulators that could drive major phenotypic changes upon SynNotch activation, limited cell types that will accept the SynNotch machinery, leaky activation, and likely others. An honest discussion of these would benefit the manuscript tremendously.

We are grateful for the reviewer's feedback and for providing guidance on how to strengthen our manuscript. We modified the manuscript to include additional discussion of the windows of operation of our technologies in the context of the larger field.

Revisions to the manuscript:

- Revised Discussion section throughout to highlight limitations related to efficiency, dynamics, etc.

4. New plasmids should be deposited on Addgene prior to acceptance and made available to the community upon publication.

We have submitted our plasmids to Addgene, and will be made available when the manuscript is published.

5. Images in Figure 4D should be substituted with those at higher resolutions, matching that presented throughout the rest of the paper.

We have updated the images in Figure 4 to those with higher resolution. Thank you for pointing this out.

Reviewers' Comments:

Reviewer #1:

Remarks to the Author:

The authors have conducted extensive new experiments to address all my comments, which is highly commendable. I now recommend the acceptance of this manuscript as is.

Reviewer #2:

Remarks to the Author:

In the revised manuscript, the authors addressed all of our concerns/suggestions from the first round of reviews. I only have one minor comment. For the quantification in panel 4D, I would appreciate it if they addressed the high proportion of BFP+ miRFP- cells in the double ligand condition. My guess is that perhaps the mCherry anti-mCherry pair is more effective at transducing the signal than the GFP anti-GFP pair. It doesn't affect the conclusion of the paper, but it could be an important consideration for people who might want to apply this synthetic ligand-receptor system in their own research.

Reviewer #3:

Remarks to the Author:

The authors have satisfactorily addressed my prior comments. I am especially appreciative to their thoughtful discussion of the systems' limitations and how it better fits in the context alternative strategies. I maintain that theirs is a beautiful study, one that I would now be happy to see published in Nature Communications.

RESPONSE TO REVIEWERS

Reviewer #1 (Remarks to the Author):

The authors have conducted extensive new experiments to address all my comments, which is highly commendable. I now recommend the acceptance of this manuscript as is.

Thanks!

Reviewer #2 (Remarks to the Author):

In the revised manuscript, the authors addressed all of our concerns/suggestions from the first round of reviews. I only have one minor comment. For the quantification in panel 4D, I would appreciate it if they addressed the high proportion of BFP+ miRFP- cells in the double ligand condition. My guess is that perhaps the mCherry anti-mCherry pair is more effective at transducing the signal than the GFP anti-GFP pair. It doesn't affect the conclusion of the paper, but it could be an important consideration for people who might want to apply this synthetic ligand-receptor system in their own research.

That was an interesting point, and we added our thoughts and comments in the results and discussion sections.

Reviewer #3 (Remarks to the Author):

The authors have satisfactorily addressed my prior comments. I am especially appreciative to their thoughtful discussion of the systems' limitations and how it better fits in the context alternative strategies. I maintain that theirs is a beautiful study, one that I would now be happy to see published in Nature Communications.

Thanks!